# DISCOVERING EVOLUTION STRATEGIES VIA META-BLACK-BOX OPTIMIZATION

**Robert Tjarko Lange**[*]
Technical University Berlin

**Tom Schaul & Yutian Chen & Tom Zahavy**
DeepMind

**Valentin Dallibard**
DeepMind

**Chris Lu**[*]
University of Oxford

**Satinder Singh & Sebastian Flennerhag**
DeepMind

## ABSTRACT

Optimizing functions without access to gradients is the remit of black-box methods such as evolution strategies. While highly general, their learning dynamics are often times heuristic and inflexible — exactly the limitations that meta-learning can address. Hence, we propose to discover effective update rules for evolution strategies via meta-learning. Concretely, our approach employs a search strategy parametrized by a self-attention-based architecture, which guarantees the update rule is invariant to the ordering of the candidate solutions. We show that meta-evolving this system on a small set of representative low-dimensional analytic optimization problems is sufficient to discover new evolution strategies capable of generalizing to unseen optimization problems, population sizes and optimization horizons. Furthermore, the same learned evolution strategy can outperform established neuroevolution baselines on supervised and continuous control tasks. As additional contributions, we ablate the individual neural network components of our method; reverse engineer the learned strategy into an explicit heuristic form, which remains highly competitive; and show that it is possible to self-referentially train an evolution strategy from scratch, with the learned update rule used to drive the outer meta-learning loop.

## 1 INTRODUCTION

Black-box optimization (BBO) methods are those general enough for the optimization of functions without access to gradient evaluations. Recently, BBO methods have shown competitive performance to gradient-based optimization, namely of control policies (Salimans et al., 2017; Such et al., 2017; Lee et al., 2022). Evolution Strategies (ES) are a class of BBO that iteratively refines the sufficient statistics of a (typically Gaussian) sampling distribution, based on the function evaluations (or fitness) of sampled candidates (population members). Their update rule is traditionally formalized by equations based on first principles (Wierstra et al., 2014; Ollivier et al., 2017), but the resulting specification is inflexible. On the other hand, the evolutionary algorithms community has proposed numerous variants of BBO, derived from very different metaphors, some of which have been shown to be equivalent (Weyland, 2010). One way to attain flexibility without having to hand-craft heuristics is to *learn* the update rules of BBO algorithms from data, in a way that makes them more adaptive and scalable. This is the approach we take: We meta-learn a neural network parametrization of a BBO update rule, on a set of representative task families, while leveraging evaluation parallelism of different BBO instances on modern accelerators, building on recent developments in learned optimization (e.g. Metz et al., 2022). This procedure discovers novel black-box optimization methods via meta-black-box optimization, and is abbreviated by *MetaBBO*. Here, we investigate one particular instance of MetaBBO and leverage it to discover a *learned evolution strategy* (LES).[1] The concrete LES architecture can be viewed as a minimal Set Transformer (Lee et al., 2019), which naturally enforces an update rule that is invariant to the ordering of candidate solutions within a batch of black-box evaluations. After meta-training, LES has learned to flexibly interpolate between copying the best-performing candidate solution (hill-climbing) and successive moving average updating (finite difference gradients). Our contributions are summarized as follows:

---

[*]Work done during an internship at DeepMind. Contact: `robert.t.lange@tu-berlin.de`.
[1]Pronounced 'meta-boh' and 'less'.

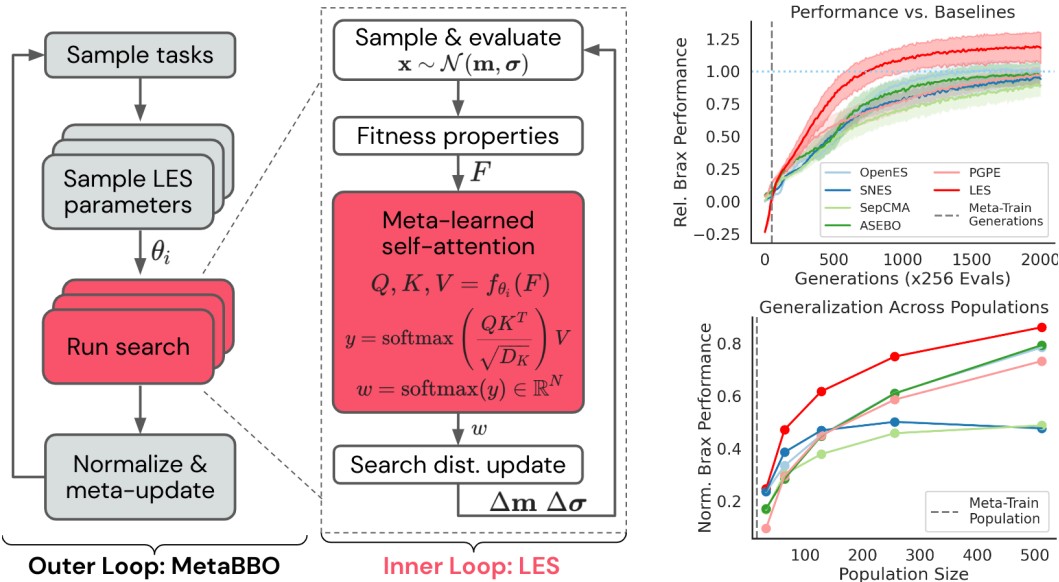

Figure 1: Overview diagram of MetaBBO, LES & performance on continuous control tasks. **Left: MetaBBO**. The outer loop samples a set of inner loop tasks (uniformly), and a set of candidate LES parameters $\{\theta_i\}_{i=1}^M$ from a meta-ES (here: CMA-ES). After obtaining a normalized meta-fitness score of each LES instance, the meta-ES is updated and we iterate the meta-optimization process. **Middle: LES**. In the inner loop and for each task, we iteratively sample candidate solutions from a Gaussian search distribution and evaluate their fitness. Afterwards, a (meta-learned) Set Transformer-inspired search strategy processes tokens of fitness transformations corresponding to the population member performance. It outputs a set of recombination weights, which are used to update the mean and standard deviation of the diagonal Gaussian search distribution. An additional MLP module (not shown) computes per-dimension learning rates by processing momentum-like statistics and a timestamp. **Right: Neuroevolution of control policies**. Average normalized performance across 8 continuous control Brax environments (Freeman et al., 2021). The LES discovered by MetaBBO on a small set of analytic meta-training problems generalizes far beyond its meta-training distribution, in terms of problem type, population size, search dimension and optimization horizon. In fact, LES outperforms diagonal ES baselines (normalized against OpenES) and scales well with an increased population size (normalized by min/max performance across all strategies). Results are averaged over 10 independent runs ($\pm1.96$ standard errors). Each task-specific learning curve is normalized by the largest and smallest fitness across all considered ES. Afterwards, the normalized learning curves are averaged across tasks. More training details and results can be found in Section 5 and in the supplementary information (SI, C and E, Figure 16).

1. We propose a novel self-attention-based ES parametrization, and demonstrate that it is possible to meta-learn black-box optimization algorithms that outperform existing hand-crafted ES algorithms on neuroevolution tasks (Figure 1, right). The learned strategy generalizes across optimization problems, compute resources and search space dimensions.

2. We investigate the importance of the meta-task distribution and meta-training protocol. We find that in order to meta-evolve a well-performing ES, only a handful of core optimization classes are needed at meta-training time. These include separable, multi-modal and high conditioning functions (Section 5).

3. We reverse-engineer the learned search strategy. More specifically, we ablate the black-box components recovering an interpretable strategy and show that all neural network components have a positive effect on the early performance of the search strategy (Section 6). The *discovered evolution strategy* provides a simple to implement yet very competitive new ES.

4. We demonstrate how to generate a new LES starting from a blank-slate LES: A randomly initialized LES can bootstrap its own learning progress and *self-referentially* meta-learn its own weights (Section 7).

## 2 RELATED WORK

**Evolution Strategies & Neuroevolution**. Unlike gradient-based optimization, ES (Rechenberg, 1973; Schwefel, 1977; Beyer & Schwefel, 2002) rely only on black-box function evaluations without the necessity of access to gradient evaluations. Instead, they iterate between the sampling of candidate solutions, their evaluation and the updating of the sampling distribution. ES can be grouped into estimation of distribution algorithms, which refine the sufficient statistics of a sampling distribution in order to increase the likelihood of high fitness solutions (Hansen & Ostermeier, 2001) and finite-difference gradient-based methods (Wierstra et al., 2014; Nesterov & Spokoiny, 2017). They can address non-differentiable problem settings and yield competitive wall-clock times on neural network tasks due to parallel fitness evaluation (Salimans et al., 2017; Such et al., 2017).

**Discovering Algorithm Components from Data**. The history of machine learning has been characterized by the successive replacement of manually designed components by modules that are informed by data. This includes the use of convolutional neural networks that learn filters (Fukushima, 1975; LeCun et al., 1998), neural architecture search (Elsken et al., 2019) or initilization schemes (Dauphin & Schoenholz, 2019). Recently, these efforts have been extended to the end-to-end discovery of objective functions in Reinforcement Learning (Kirsch et al., 2019; Oh et al., 2020; Xu et al., 2020; Lu et al., 2022), the online refinement of hyperparameter schedules (Xu et al., 2018; Zahavy et al., 2020; Flennerhag et al., 2021; Parker-Holder et al., 2022) and the meta-learning of entire learning algorithms (Wang et al., 2016; Kirsch & Schmidhuber, 2021; Kirsch et al., 2022).

**Meta-Learned Gradient-Based Optimization**. An exciting direction aims to meta-learn gradient descent-based learning rules (Bengio et al., 1992; Andrychowicz et al., 2016; Metz et al., 2019) to update the weights of a neural network. Unlike our proposed method, these approaches require access to gradient calculations using the backpropagation algorithm implemented by automatic differentiation software. The gradient and several summary statistics are processed by a small neural network, whose weights have been meta-learned on a distribution of tasks (Metz et al., 2020). The resulting outputs are calculated on a per-parameter basis and used to characterize the magnitude and direction of the weight change. It has been shown that this approach can extract useful inductive biases for domain-specific gradient-based optimization (Merchant et al., 2021; Metz et al., 2022) but can struggle to broadly generalize to task domains beyond the meta-training distribution.

**Meta-Learned Black-Box Optimization**. Shala et al. (2020) meta-learn a policy that controls the scalar search scale of CMA-ES (Hansen & Ostermeier, 2001). Chen et al. (2017); TV et al. (2019); Gomes et al. (2021) explored meta-learning entire algorithms for low-dimensional BBO. They parametrize the algorithm by a recurrent network processing the raw solution vectors and/or their associated fitness. Their applicability is constrained to narrow optimization domains, fixed population sizes or search dimensions. The proposed LES, on the other hand, learns to recombine solution candidates by leveraging the invariance properties of dot-product self-attention (Lee et al., 2019; Tang & Ha, 2021). After meta-training the learned ES generalizes to unseen populations and a large number of dimensions ($> 5000$). To the best of our knowledge we are the first to demonstrate that a meta-learned ES generalizes to neuroevolution tasks. Our meta-evolution approach does not rely on access to a teacher ES (Shala et al., 2020) or knowledge of task optima (TV et al., 2019).

**Self-Referential Meta-Learning**. Our current learning systems are not capable of flexibly refining their own learning behavior. Schmidhuber (1987) first articulated the vision of self-referentially updating a genetic algorithm to improve its own performance. Kirsch & Schmidhuber (2022) further formulated a simple heuristic to allocate compute resources for population-based self-referential learning. Metz et al. (2021a) showed that the recursive meta-training of gradient-based optimizers is feasible using an improvement mechanism based on population-based training (Jaderberg et al., 2017). Here, we introduce a simple self-referential MetaBBO loop, which successfully meta-evolves a LES capable of generalization to continuous control tasks.

## 3 PRELIMINARIES: GAUSSIAN EVOLUTION STRATEGIES

Many common ES use Gaussian search distributions, with the most scalable ones using isotropic or diagonal (axis-aligned) Gaussian approximations. They have shown to effectively scale to large search space dimensions, specifically in the context of neuroevolution (Salimans et al., 2017), and can outperform full covariance search methods (Ros & Hansen, 2008). We therefore chose them as the starting point for our attention-based LES architecture, and use four common ones as baselines:

OpenES (Salimans et al., 2017), sep-CMA-ES (Ros & Hansen, 2008), PGPE (Sehnke et al., 2010) and SNES (Schaul et al., 2011). We also compare with ASEBO (Choromanski et al., 2019), which dynamically adapts to the fitness geometry based on the iterative estimation of gradient subspaces.

Diagonal ES maintain mean $\mathbf{m} \in \mathbb{R}^D$ and standard deviation vectors $\boldsymbol{\sigma} \in \mathbb{R}^D$, where $D$ denotes the number of search space dimensions. At each generation one samples a population of candidate solutions $\mathbf{x}_j \in \mathbb{R}^D \sim \mathcal{N}(\mathbf{m}, \boldsymbol{\sigma} \mathbf{1}_D)$ for all $j = 1, ..., N$ population members. Each solution vector is evaluated, producing fitness estimates $f_j \in \mathbb{R}$, which are used to construct recombination weights $\boldsymbol{w}_j$. These weights define how to update the search distribution (mean and standard deviations). Most often, the weights are assigned based on the rank of the fitness $\mathrm{rank}(j) \in [1, N]$ among the current population, for example:

$$w_{t,j}(\{f_k\}_{k=1}^N) = w_{\mathrm{rank}(j)} = \begin{cases} \frac{1}{E}, & \text{if } \mathrm{rank}(j) \leq E, \\ 0, & \text{otherwise}, \end{cases} \tag{1}$$

where $w_{\mathrm{rank}(j)}$ denotes the weight assigned to the rank of member $j$, and $E$ denotes the number of high-fitness candidates that contribute to the update. The search distribution is then updated using an exponentially moving fitness-weighted average:

$$\mathbf{m}_{t+1} = (1 - \boxed{\boldsymbol{\alpha}_{m,t}})\mathbf{m}_t + \boxed{\boldsymbol{\alpha}_{m,t}} \sum_{j=1}^N \boxed{\boldsymbol{w}_{t,j}} \mathbf{x}_j, \tag{2}$$

$$\boldsymbol{\sigma}_{t+1} = (1 - \boxed{\boldsymbol{\alpha}_{\sigma,t}})\boldsymbol{\sigma}_t + \boxed{\boldsymbol{\alpha}_{\sigma,t}} \sqrt{\sum_{j=1}^N \boxed{\boldsymbol{w}_{t,j}} (\mathbf{x}_j - \mathbf{m}_t)^2}, \tag{3}$$

which can be interpreted as a finite-difference gradient update on the expected fitness. Both the learning rates $\boldsymbol{\alpha}_m, \boldsymbol{\alpha}_\sigma$ and the weighting scheme $\boldsymbol{w}_t$, highlighted in red, are commonly fixed across the update generations $t = 1, ..., T$, which is an obvious restriction to the flexibility of the ES. The main aim of our proposed meta-learned neural network parametrization is to overcome the implied limitations by meta-learning these.

## 4 META-EVOLVING EVOLUTION STRATEGIES

In this section, we first introduce the learned evolution strategy architecture (Figure 1, middle), a self-attention-based generalization of the diagonal Gaussian ES in equations 2 and 3. Section 4.2 then outlines the MetaBBO procedure (Figure 1, left) used to meta-evolve the weights of LES.

### 4.1 *Learned Evolution Strategies (LES)*: POPULATION ORDER-INVARIANCE VIA ATTENTION

A fundamental property that any (learned) ES has to fulfill is invariance in the ordering of population members within a generation. Intuitively, the order of the population members is arbitrary and should therefore not affect the search distribution update. A natural inductive bias for an appropriate neural network-based parametrization is given by the dot-product self-attention mechanism. Our proposed learned evolution strategy processes a matrix $F_t$ of population member-specific tokens (Figure 1, middle). $F_t \in \mathbb{R}^{N \times 3}$ consists of (1) the $z$-scored population fitness scores, (2) a centered rank transformation (lying within $[-0.5, 0.5]$), and (3) a Boolean indicating whether the fitness score exceeds the previously best score. This fitness matrix is embedded into queries ($Q \in \mathbb{R}^{N \times D_K}$), keys ($K \in \mathbb{R}^{N \times D_K}$) and values ($V \in \mathbb{R}^{N \times 1}$) using learned linear transforms with weights $W_K, W_Q, W_V$. Recombination weights are computed as attention scores over the members:

$$\boldsymbol{w}_t = \mathrm{softmax}\left(\mathrm{softmax}\left(\frac{QK^T}{\sqrt{D_K}}\right)V\right) = \mathrm{softmax}\left(\mathrm{softmax}\left(\frac{F_t W_Q W_K^T F_t^T}{\sqrt{D_K}}\right)F_t W_V\right) \in \mathbb{R}^N.$$

The resulting recombination weights are shared across the search space dimensions, but vary across the generations $t$. Instead of using a set of fixed learning rates, we additionally modulate $\boldsymbol{\alpha}_{m,t}, \boldsymbol{\alpha}_{\sigma,t} \in \mathbb{R}^D$ by a MLP parametrized by $\phi$ and shared across $D$. It processes a timestamp embedding and parameter-specific information provided by momentum-like statistics (see SI A). The chosen parametrization can in principle smoothly interpolate between integrating information from many population members and copying over a single best performing solution by setting a single

recombination weight and $\boldsymbol{\alpha}_{m,t}$ to one for all dimensions. The set of LES parameters are given by $\theta = \{W_K, W_Q, W_V, \phi\}$, whose total number is usually small ($<500$) given a sufficient key embedding size ($D_K = 8$). The complexity of the proposed LES scales quadratically in the population size, which for common application budgets ($N < 10,000$) introduces a negligible runtime cost.

## 4.2  *MetaBBO*: Meta-Black-Box Optimization of Black-Box Optimizers

How do we learn the LES parameters $\theta$, so that they provide useful inductive biases for black-box optimization? Meta-learning via evolutionary optimization has previously proven successful in the context of learned gradient-based optimization (Vicol et al., 2021) and RL objective discovery (Lu et al., 2022). It avoids exploding meta-gradients (Metz et al., 2021b), myopic meta-solutions (Flennerhag et al., 2021; Lange & Sprekeler, 2022), and allows to optimize $\theta$ through long inner loop ES executions involving stochastic sampling. We adopt this meta-evolution approach in MetaBBO and iterate the following steps (Figure 1, left; SI Algorithm 2):

1. **Meta-sampling**. At each meta-generation we sample a population of LES network parametrizations $\theta_i$ for $i = 1, \ldots, M$ meta-population members using a standard off-the-shelf ES algorithm with meta-mean $\mu$ and covariance $\Sigma$.

2. **Inner loop search**. Next, we estimate the performance of different LES parametrizations on a set of tasks. We sample a set of inner loop optimization problems. For each task we initialize a mean $\mathbf{m}_0 \in \mathbb{R}^D$, standard deviation $\boldsymbol{\sigma}_0 \in \mathbb{R}^D$. Each LES instance is then executed on a batch of inner loop tasks (in parallel) with $N$ population members and for a fixed set of inner-loop generations $t = 1, \ldots, T$.

3. **Meta-normalize**. In order to ensure stable meta-optimization with optimization problems that can exhibit very different fitness scales, we normalize ($z$-score) the fitness scores within tasks and across meta-population members. We average the scores over inner-loop tasks, and maximize over inner-loop population members and over generations.

4. **Meta-updating**. Given the meta-performance estimate for the different LES parametrizations, we update the meta-ES search distribution $\mu', \Sigma'$ and iterate.

The choice of the meta-task distribution is crucial to ensure generalization of the meta-learned evolution strategy. We select a small set of classic black-box optimization functions, to characterize a space of representative optimization problems. The problem space includes smooth functions with and without high curvature and multi-modal surfaces with global and local structure. Furthermore, we apply Gaussian additive noise to the fitness evaluations to mimic unreliable estimation and sample optima offsets. MetaBBO differs from standard meta-learning in a couple aspects: First, we use BBO in both the inner and outer loop. This allows for flexible LES design, stochastic sampling and long inner loop ES execution. We also share the same task parameters, $\mathbf{m}_0$ and fixed randomness for all meta-population members $\theta_i$. The fixed stochasticity and task-based normalization enhances stable meta-optimization ('Pegasus-trick', Ng & Jordan, 2000).

## 5  Experiments

In this section we discuss the results of meta-learning LES parameters via MetaBBO. We answer the following questions: Does MetaBBO discover an ES capable of generalization beyond the meta-training distribution and can the LES meta-trained only on a limited set of functions even generalize to unseen neural network tasks with different fitness functions & network architectures?

### 5.1  Meta-Training on the Black-Box Optimization Benchmark

Throughout, we meta-train on functions from the BBOB (Hansen et al., 2010) benchmark, which comprise a set of challenging functions for optimization (Table 1). From BBOB, we construct a *small*, *medium*, and *large* meta-training set (detailed in Section 5.2). To create a task instance, we randomly sample: (1) a function $f$ from the meta-training set, (2) a dimensionality $D \sim U\{2, 3, \ldots, 10\}$ for $f : \mathbb{R}^D \to \mathbb{R}$, (3) a noise-level $\Xi \sim U[0, 0.1]$, and (4) an optimum offset $\boldsymbol{z} \sim U[-5, 5]^D$, which together creates a stochastic objective $\hat{f}(\boldsymbol{x}) \sim f(\boldsymbol{x} + \boldsymbol{z}) + \Xi \mathcal{N}(0, 1)$. Finally, (5) we sample an initialization $\boldsymbol{m}_0 \sim [-5, 5]^D$ and starting generation counter $t_0 \in [0, 2000]$.

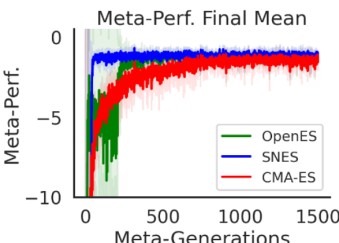 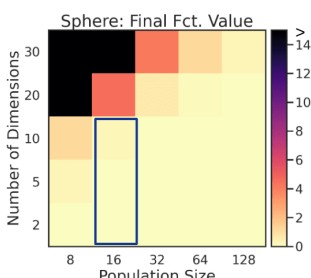 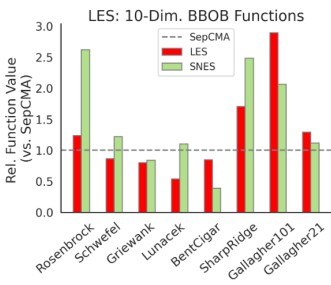

Figure 2: BBOB evaluation of LES. **Left**. MetaBBO fitness objective over meta-generations. The optimization process is robust to the choice of meta-ES. **Middle**. Minimization performance on Sphere task across dimensions and population size ($T = 25$). Boxed cells indicate meta-training tasks. **Right**. Relative performance on different BBOB test functions ($D = 10, N = 16, T = 100$), lower is better. Despite LES only being meta-trained on a population size of $N = 16$ and low-dimensional ($D \leq 10$) search spaces, it is capable of generalizing to different functions, problem settings and resources. Detailed meta-training curves can be found in SI E (Figures 8, 9).

In the MetaBBO outer loop we optimize $\theta$ using CMA-ES (Hansen & Ostermeier, 2001) for 1500 meta-generations with a meta-population size of $M = 256$. We sample 128 tasks and train each sampled LES on each task independently. Each LES is unrolled for $T = 50$ generations, with $N = 16$ population members in each iteration.[2] Each LES-task pair is assigned a raw fitness score according to the highest fitness (lowest loss) obtained in the rollout. These raw scores are meta-normalized per task before we aggregate them into a meta-fitness score for each sampled LES. During MetaBBO training, we find average meta-training improves rapidly (Figure 2, top left) and is invariant to the choice of the meta-ES optimizer. We assess the performance of LES obtained through the MetaBBO outer loop, on the BBOB functions (Figure 2, right). LES generalizes well to larger dimensions and larger evaluation budgets on both meta-training and meta-testing function classes. While the task complexity increases with $D$ and performance drops for a fixed population size, LES is capable of making use of the additional evaluations provided by an increased population.

## 5.2 META-GENERALIZATION OF LES TO UNSEEN NEUROEVOLUTION TASKS

Our results on BBOB suggests that LES have strong generalizing properties, thus prompting the question, what are the limits of LES' generalization capabilities? In this section, we evaluate an LES trained on BBOB on a set of CNN-based classification tasks and continuous control tasks (Figure 3) using MLP policies. This requires LES to generalize beyond the 10-dimensional problems it was trained on to thousands of dimensions. Not only that, but also to completely different loss surfaces, and even learning paradigms (in the case of continuous control).

Figure 3 shows that LES is indeed capable of successfully evolving CNN classifiers on different MNIST variants. We also find that it is capable of evolving 4-layer MLP policies for robotics tasks (Freeman et al., 2021). We consider three budgets for *test-time* optimisation (that vary in inner loop length $T$ and population size $N$, see Figure 3). LES generally outperforms ES baselines[3] across test-time budgets. It enjoys a substantial advantage in small budgets, where it outperforms baselines on 5 out of 7 tasks. LES still performs best on the majority of tasks for the medium and large adaptation budgets, which requires it to generalize to larger populations and longer unroll lengths. Next, we evaluate the impact of the meta-training distribution on the generalization of LES. The *small* meta-training set consists of 1 BBOB function and restricts dimensionality to $D = 2$ with inner loop length $T = 25$. *medium* is restricted to 5 functions., with $D \in \{1, 2, \ldots, 5\}$ and $T = 25$; *large* is given by 10 functions, $D \in \{1, 2, \ldots, 10\}$ and $T = 50$. Figure 4 shows that training on broader meta-training distributions has a positive effect on the generalization capability of the trained LES, as measured on continuous control tasks. In particular, we find that meta-training with larger populations in the inner loop especially benefits from scaling up the meta-training set.

---

[2]In SI Figures 10, 11, we demonstrate that the MetaBBO is largely robust to the choice of $D, T, N, D_K$.

[3]For all considered ES we grid-searched over different $\boldsymbol{\sigma}_0$ and tuned learning rates were additionally tuned on the Brax ant environment. More details can be found in the SI D. We provide more results on 4 MinAtar (Young & Tian, 2019) tasks with CNN policies in the SI (Figure 21).

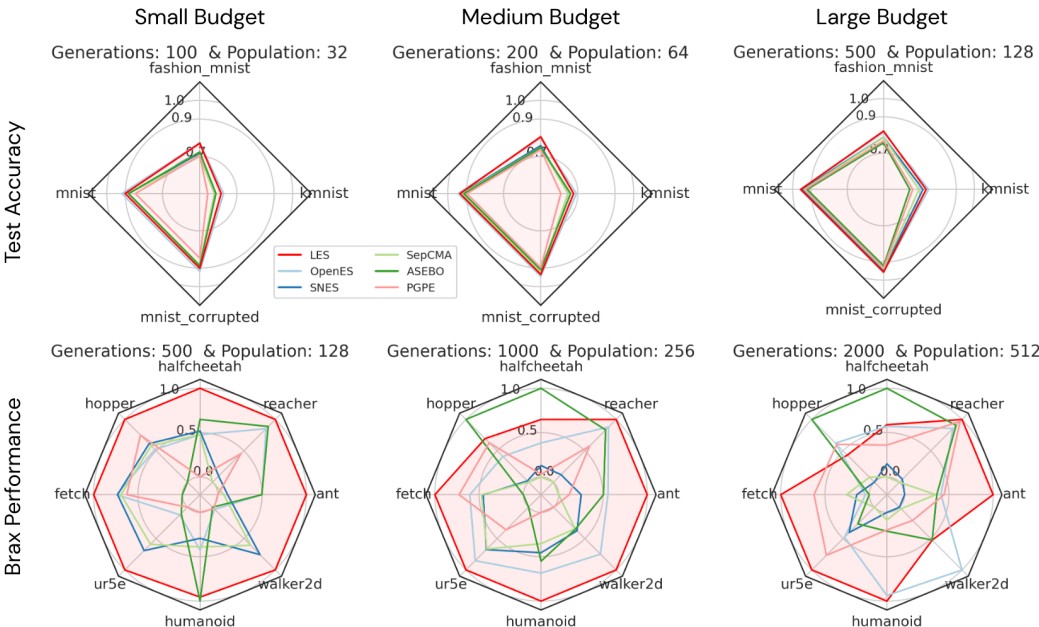

Figure 3: Evaluation of LES on classification and continuous control tasks. **Top**. BBOB meta-evolved LES generalizes to classification tasks including MNIST/Fashion-MNIST/K-MNIST classification using a simple Convolutional Neural Network. **Bottom**. The learned strategy can also evolve control policies based on a 4 hidden layer Tanh MLP architecture with 32 hidden units in 8 different control environments. Results are averaged over 5 (vision) and 10 (control) independent runs and max/min normalized for Brax for better comparison. Details: SI E (Figures 16, 20).

## 6    REVERSE ENGINEERING LEARNED EVOLUTION STRATEGIES

In order to investigate which algorithmic components are discovered by the MetaBBO procedure, we start by ablating the different meta-learned neural network modules. Fixing both recombination weights and the learning rate modulation, as per the simple Gaussian ES in Section 3 degrades performance significantly (Figure 5, top left). Fixing the recombination weights (as in equation 1, $E = 0.5N$) also decreases performance, but fixing the learning rates to $\alpha_m = 1, \alpha_\sigma = 0.1$ does not hurt performance significantly. We observed that the learning rates are flexibly adapted early in training, but settle quickly to constant values (Figure 5, right). Next, we visualized the recombination weights for a LES run on the Walker2D Brax task (Figure 5, top middle). LES performs a variable amount of selection and either applies hill-climbing or integrates information

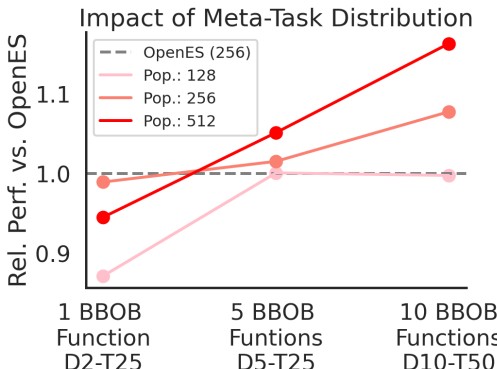

Figure 4: The performance of LES increases as it is meta-trained on richer task distributions. We plot the relative performance of LES on Brax tasks against OpenES (Salimans et al., 2017).

from multiple well-performing solutions. We plot the weights as a function of the centered rank (for $N = 256$) for sampled Gaussian fitness scores (Figure 5, bottom middle). The recombination weights can be well approximated by a softmax parametrization of the centered ranks:

$$\boldsymbol{w}_{t,j} = \text{softmax}\left(20 \times \left[1 - \text{sigmoid}\left(\beta \times (\text{rank}(j)/N - 0.5)\right)\right]\right), \ \forall j = 1, ..., N, \qquad (4)$$

with a temperature parameter $\beta \approx 12.5$. Intuitively, the temperature regulates the amount of selection pressure applied by the recombination weights. Based on these insights, we implement an interpretable ES that uses the discovered recombination weight approximation. We call the resulting

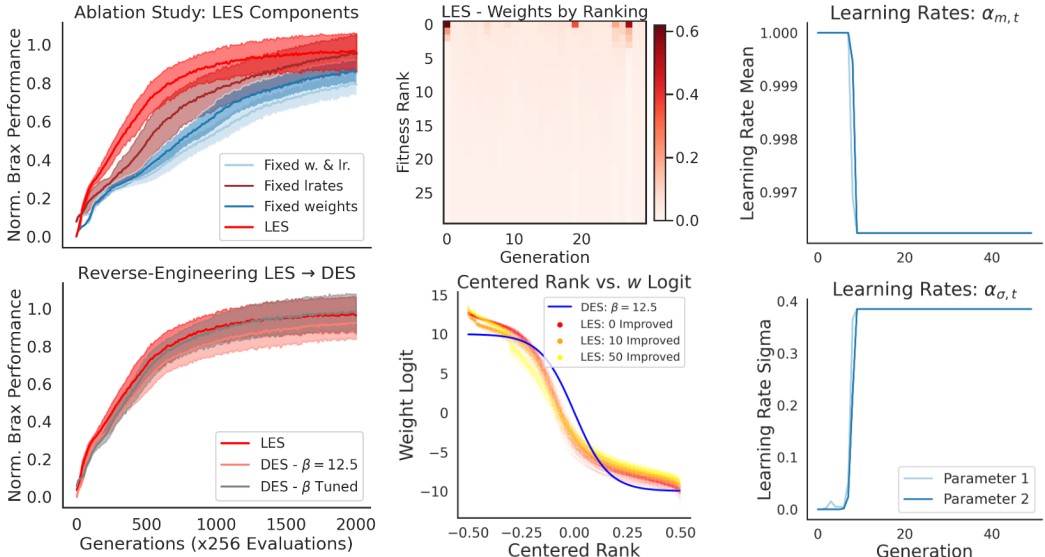

Figure 5: Reverse engineering of LES. **Left**. Ablation study of LES and performance of reverse-engineered DES on continuous control tasks. The self-attention-based recombination weights contribute most to the performance of LES. Furthermore, DES provides a competitive ES when sufficiently tuning $\beta$. Results are averaged over 5 independent runs ($\pm 1.96$ standard errors). **Middle**. Recombination weights in the first 20 LES generations on the Walker2D task. LES can perform both hill climbing by assigning high recombination weight to the best performing population member and distribute the recombination weight across multiple well-performing candidates. The LES recombination weights can be approximated well by equation (4). **Right**. Learning rate modulation of LES on a subset of dimensions. LES learning rates quickly converge to fixed values after initial adaption. Detailed learning curves can be found in SI E (Figures 13, 17).

algorithm *Discovered Evolution Strategy* (DES, SI Algorithm 1). DES($\beta = 12.5$) performs similarly to LES on the Brax benchmark tasks and can be further improved by tuning the temperature parameter (Figure 5, bottom left). This result highlights that MetaBBO can be used not only to discover black-box ES strategies, but also to discover new interpretable inductive biases.

# 7 SELF-REFERENTIAL META-EVOLUTION OF LES

Given a choice of meta-ES, MetaBBO can discover LES that performs competitively with baseline ES. It is therefore natural to ask, can we use an LES to discover a new LES? And can this be a purely self-referential loop, starting from a random initialization? To address this question we consider a two-step self-referential metaBBO paradigm, which starts from a random $\theta_{meta}$ (Figure 6, left):

1. We evaluate a set of candidate LES parametrizations $\theta_i \sim \mathcal{N}(\mu, \Sigma)$ with $\Sigma = \boldsymbol{\sigma}\mathbb{1}$. After obtaining the meta-fitness scores $f(\theta_i)$, we update the strategy statistics using the meta-LES update rule $\mu', \Sigma' \leftarrow \text{LES}(\{\theta_i, f_i\}_{i=1}^M | \mu, \Sigma, \theta_{meta})$.

2. If the meta-fitness has improved, we replace $\theta_{meta} \leftarrow \arg\max_{\theta_i} f(\theta_i)$ using a hill-climbing step that copies the best performing inner-loop mean $\mu' \leftarrow \theta_{meta}$. Additionally, we reset the meta-search variance to induce additional exploration around the newly improved meta-search mean: $\Sigma \leftarrow \sigma_0 \mathbb{1}_{D_{meta}}$.

Note that this procedure starts out as simple random search with hill-climbing selection. As the meta-parameters are updated, the meta-search is improved by the newly learned inductive biases. Thereby, LES is capable of improving its own optimization behavior. Figure 6 compares different meta-training procedures for LES, including the usage of CMA-ES, a previously meta-trained and fixed LES instance, and our proposed self-referential training paradigm. We find it is indeed feasible to effectively self-evolve LES starting from a random initialization. While the meta-fitness is improved fastest by the pre-trained LES instance, we observe that the resulting meta-trained LES

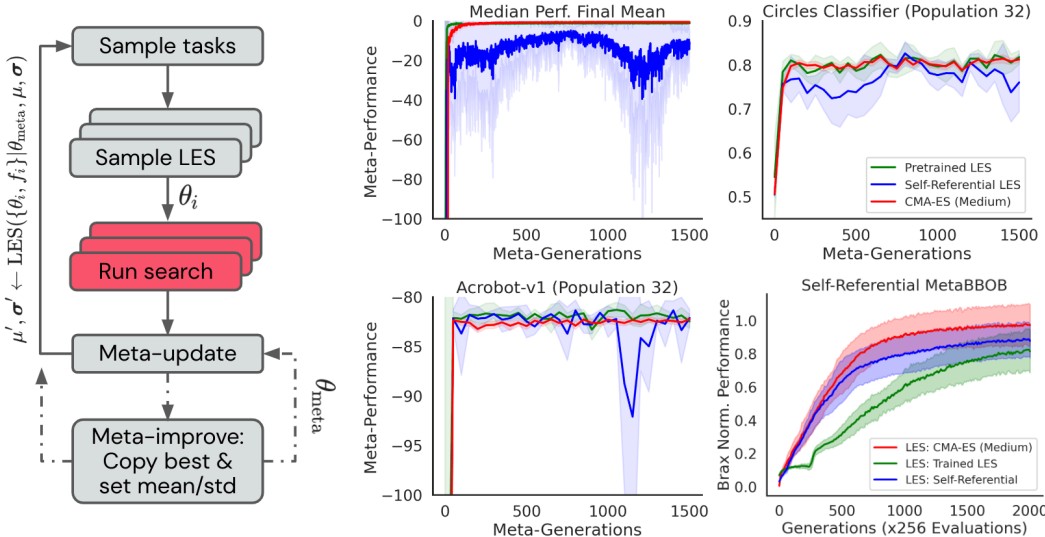

Figure 6: Self-Referential MetaBBO of LES. **Left**. We use LES to meta-evolve its own parameters $\theta$ starting from a random initialization and using hill climbing replacement. **Middle**. Meta-performance over training. It is possible to meta-train LES using both a previously trained and frozen instance and a self-referentially improved random initialization. **Right, Top**. Performance on 2D Circles MLP classifier and Acrobot control tasks throughout meta-training. **Right, Bottom**. While the LES discovered by a frozen pre-trained version does not generalize to Brax meta-testing, the self-referentially meta-trained one does. Results are averaged over 5 independent runs (+/- 1.96 standard errors). Detailed learning curves can be found in SI E (Figures 18, 9).

does not generalize well to the downstream Brax evaluation (Figure 6, bottom right). Both the CMA-ES optimizer and self-referential meta-training procedure, on the other hand, are capable of meta-evolving stable and performant LES, that do not overfit to the meta-training task distribution. We speculate that self-referential meta-training performs implicit regularization: LES is forced to simultaneously perform well on the inner loop tasks and to progress its own continual improvement.

# 8 CONCLUSION & DISCUSSION

**Summary.** We introduced a self-attention-based ES architecture and a procedure to effectively meta-learn its parametrization. The resulting learned evolution strategy is flexible and outperforms established baseline strategies on both vision and continuous control tasks. We reverse engineered the discovered search update rule into a simple and competitive DES. Finally, we showed that it is possible to successfully self-referentially evolve a LES starting from a random initialization of itself.

**Limitations.** The LES considered in this paper only applies to diagonal covariance strategies. This limits its effectiveness on non-separable problems. We observed that the learning rate module can struggle to generalize to long time horizons indicating a degree of temporal meta-overfitting. Finally, the self-attention mechanism scales quadratically in the number of population members. This may be addressed by improvements in Transformer efficiency (Wang et al., 2020; Jaegle et al., 2021).

**Future work.** Our LES architecture meta-learns to modulate the search distribution update solely based on fitness transformations. One potential extension is to provide additional information (agent trajectory or multiple evaluations) to the attention module. Thereby, the MetaBBO could discover behavioural descriptors in the context of quality-diversity optimization (Pugh et al., 2016; Cully & Demiris, 2017) or application-specific denoising procedures. Furthermore, one may want to attend over an archive of well-performing solutions (e.g. as in genetic algorithms) using cross-attention that implements a cross-over operation. Finally, our experiments indicate the potential of further scaling of the MetaBBO procedure to broader meta-training task distributions.

## ACKNOWLEDGEMENTS

Work funded by DeepMind. We thank Nemanja Rakićević for valuable feedback on a first draft of this manuscript. Furthermore, the authors are grateful to Antonio Orvieto, Denizalp Goktas, Evgenii Nikishin, Junhyok Oh, Greg Farquhar, Alexander Neitz, Himanshu Sahni, Ted Moskovitz and other colleagues on the Discovery team and at DeepMind for stimulating discussions over the course of the project.

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

# Appendix

## Table of Contents

## A  META-TRAINING DETAILS

| Function Name | Reference | Property | Meta-Distribution |
|---|---|---|---|
| Sphere | Hansen et al. (p. 5, 2010) | Separable (Independent) | Small |
| Rosenbrock | Hansen et al. (p. 40, 2010) | Moderate Condition | Medium |
| Discus | Hansen et al. (p. 55, 2010) | High Condition | Medium |
| Rastrigin | Hansen et al. (p. 75, 2010) | Multi-Modal (Local) | Medium |
| Schwefel | Hansen et al. (p. 100, 2010) | Multi-Modal (Global) | Medium |
| BuecheRastrigin | Hansen et al. (p. 20, 2010) | Separable (Independent) | Large |
| AttractiveSector | Hansen et al. (p. 30, 2010) | Moderate Condition | Large |
| Weierstrass | Hansen et al. (p. 80, 2010) | Multi-Modal (Global) | Large |
| SchaffersF7 | Hansen et al. (p. 85, 2010) | Multi-Modal (Global) | Large |
| GriewankRosenbrock | Hansen et al. (p. 95, 2010) | Multi-Modal (Global) | Large |

Table 1: Meta-training black-box optimization tasks with different structural properties.

During meta-training we uniformly sample task parameters $\xi_k$ for $k = 1, ..., K$ different tasks, characterizing the inner fitness $f : \mathbb{R}^D \to \mathbb{R}$, namely the function identifier and the number of dimensions $D \in [2, 10]$. In addition, we sample a constant offset to the optimum $x^\star$ and a noise level $\mathbb{E}_\epsilon[f(x) + \epsilon] = f(x)$ with $\epsilon \sim \mathcal{N}(0, \Xi)$. The initial mean is sampled $\mathbf{m}_0 \in [-5, 5]^D$ in order to ensure solutions that are robust to their starting initialization. We unroll the LES for $T = 50$ generations with $N = 16$ population members/function evaluations each. The meta-objective is computed based on the collected inner loop fitness scores:

$$\left[ \left[ \left[ \{f(x_{j,t}|\xi_k)\}_{j=1}^N \right]_{t=1}^T \right]_{k=1}^K |\theta_i \right]_{i=1}^M \in \mathbb{R}^{N \times T \times K \times M}.$$

For each candidate LES $\theta_i$ we minimize over time and population members. Afterwards, we $z$-score the task-specific results over all population members and compute the mean across all tasks:

$$\left[[f(\theta_i|\xi_k)]_{i=1}^M\right]_{k=1}^K = \min_t \min_j \left[\left[\left[\{f(x_{j,t}|\xi_k)\}_{j=1}^N\right]_{t=1}^T\right]_{k=1}^K |\theta_i\right]_{i=1}^M$$

$$\{\tilde{f}(\theta_i)\}_{i=1}^M = \texttt{Median}\left[\texttt{Z-Score}\left([f(\theta_i|\xi_k)]_{i=1}^M\right)\right]_{k=1}^K.$$

In practice we found that it can be helpful to meta-optimize the median across tasks of the $z$-scored fitness. Inspired by Shala et al. (2020), the learning rate modulation of $\boldsymbol{\alpha}_m, \boldsymbol{\alpha}_\sigma$ is performed using a small MLP that outputs two sigmoid activations corresponding to the per-parameter learning rates (mean/std) using a tanh timestamp embedding as in (Metz et al., 2022) and a set of evolution paths, which mimic momentum terms at different timescales:

$$\boldsymbol{\alpha}_{m,t,d}, \boldsymbol{\alpha}_{\sigma,t,d} = \text{MLP}(\mathbf{p}_{c,t+1,d}, \mathbf{p}_{\sigma,t,d}, \rho(t))$$

$$\mathbf{p}_{c,t+1} = (1 - \boldsymbol{\alpha}_{p_c})\mathbf{p}_{c,t} + \boldsymbol{\alpha}_{p_c}\left(\sum_j w_j(x_j - \mathbf{m}_t) - \mathbf{p}_{c,t}\right) \in \mathbb{R}^{D \times 3}$$

$$\mathbf{p}_{\sigma,t+1} = (1 - \boldsymbol{\alpha}_{p_\sigma})\mathbf{p}_{\sigma,t} + \boldsymbol{\alpha}_{p_\sigma}\left(\sum_j w_j(x_j - \mathbf{m}_t)/\boldsymbol{\sigma}_t - \mathbf{p}_{\sigma,t}\right) \in \mathbb{R}^{D \times 3},$$

with three time-scales $\boldsymbol{\alpha}_{p_c} = \boldsymbol{\alpha}_{p_\sigma} = [0.1, 0.5, 0.9]$ and $\rho(t) = \tanh(t/\gamma - 1)$ with $\gamma = [1, 3, 10, 30, 50, 100, 250, 500, 750, 1000, 1250, 1500, 2000]$. The 'medium' set of BBOB meta-training functions for $D = 2$ and the time embedding features are visualized in figure 7:

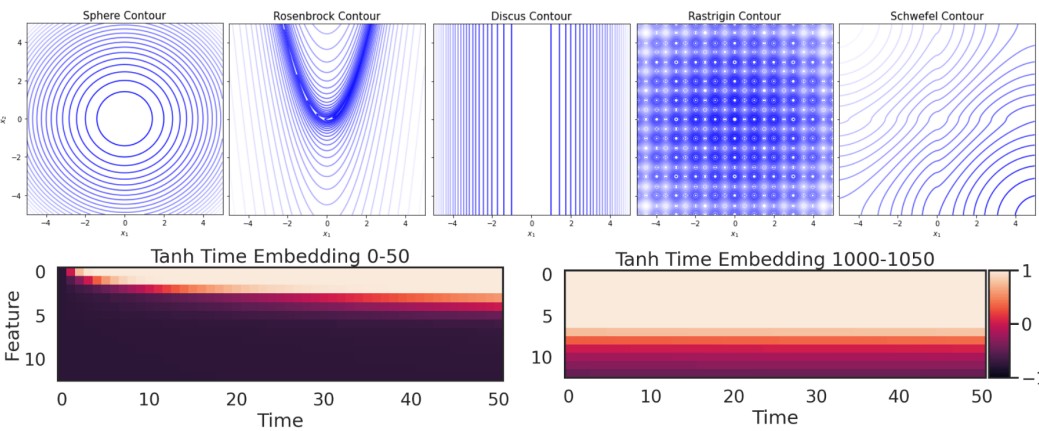

Figure 7: Visualization of 2D 'medium' BBOB functions and tanh timestamp embedding features.

## B DISCOVERED EVOLUTION STRATEGY

---

**Algorithm 1** Discovered Evolution Strategy (DES)

---

1: **Inputs:** Population members $N$, search dimensions $D$, generations $T$, Initial scale $\boldsymbol{\sigma}_0 \in \mathbb{R}^D$, Mean and std learning rates $\alpha_m = 1, \alpha_\sigma = 0.1$, temperature parameter $\beta \approx 12.5$.
2: Initialize the search mean $\mathbf{m}_0 = \mathbf{0}_D \in \mathbb{R}^D$
3: **for** $t = 0, ..., T - 1$ **do**
4:     Sample candidates: $x_j \sim \mathcal{N}(\mathbf{m}_t, \sigma_t \mathbb{1}_D)$
5:     Evaluate all candidates: $f_j, \ \forall j = 1, ..., N$
6:     Compute weights using: $\boldsymbol{w}_{t,j} = \text{softmax}\left(20 \times \text{sigmoid}\left(\beta \times (\text{rank}(j)/N - 0.5)\right)\right), \ \forall j = 1, ..., N$
7:     Update mean using equation (2) $\rightarrow \mathbf{m}_{t+1}$
8:     Update std using equation (3) $\rightarrow \boldsymbol{\sigma}_{t+1}$
9: **end for**

---

## C  METABBO ALGORITHM FOR META-EVOLVING EVOLUTION STRATEGIES

---

**Algorithm 2** MetaBBO Training of Learned Evolution Strategies

---

1: **Inputs:** Meta-population members $M$, meta-task size $K$, inner loop population size $N$, inner loop generations $T$, MetaES (e.g. CMA-ES)
2: Initialize the meta-search mean and covariance $\mu, \Sigma$.
3: **while** not done **do**
4:     Sample $K$ tasks with parameters $\xi_k, \ \forall k = 1, ..., K$
5:     Sample LES candidates: $\theta_i \sim \mathcal{N}(\mu, \Sigma), \ \forall i = 1, ..., M$
6:     Evaluate all $M$ LES candidates (in parallel) on $K$ tasks:
7:     **for** $k = 1, ..., K$ **do**
8:         **for** $i = 1, ..., M$ **do**
9:             Initialize inner loop search means/std $\mathbf{m}_{0,k,i} \in \mathbb{R}^{D_k}, \boldsymbol{\sigma}_{0,k,i} \in \mathbb{R}^{D_k}$
10:            **for** $t = 0, ..., T - 1$ **do**
11:                Sample inner loop candidates: $x_{j,t} \sim \mathcal{N}(\mathbf{m}_{t,k,i}, \boldsymbol{\sigma}_{t,k,i}\mathbf{1}_{D_k}), \ \forall j = 1, ..., N$
12:                Evaluate all inner loop candidates: $f(x_{j,t}|\xi_k), \ \forall j = 1, ..., N$
13:                Update LES mean/std using equation (2) and (3) $\to \mathbf{m}_{t+1,k,i}, \boldsymbol{\sigma}_{t+1,k,i}$ with $\theta_i$
14:            **end for**
15:        **end for**
16:    **end for**
17:    Collect all inner loop fitness scores $\left[\left[\left[\{f(x_{j,t}|\xi_k)\}_{j=1}^{N}\right]_{t=1}^{T}\right]_{k=1}^{K}|\theta_i\right]_{i=1}^{M} \in \mathbb{R}^{N \times T \times K \times M}$
18:    Compute the population normalized and task aggregated meta-fitness scores $\{\tilde{f}(\theta_i)\}_{i=1}^{M}$
19:    Update meta-mean, meta-std $\mu', \Sigma' \gets \texttt{MetaES}(\{\theta_i, \tilde{f}\}_{i=1}^{M}|\mu, \Sigma)$
20: **end while**

---

## D  HYPERPARAMETER SETTINGS

### D.1  METABBO HYPERPARAMETERS

| MetaBBO Hyperparameters for LES Discovery | | | |
|---|---|---|---|
| Hyperparameter | Value | Hyperparameter | Value |
| Meta-Generations | 1500 | Meta-Population | 256 |
| Meta-Tasks | 128 | Meta-ES | CMA-ES |
| CMA-ES $\sigma_0$ | 0.1 | Inner-Population | 16 |
| Inner-Generations | 50 | $\boldsymbol{m}_0$ range | [-5, 5] |
| Timestamp Range | [0, 2000] | $\boldsymbol{\sigma}_0$ range | [1, 1] |
| $D_k$ Attention keys | 8 | MLP Hidden dim. | 8 |

For the self-referential meta-training results (Section 7 we replace the meta-LES parameters every 5 meta-generations if we observe an improvement in the meta-objective by one of the inner LES candidates. Furthermore, we reset the meta-variance $\Sigma = \sigma_0 I = 0.1I$ .

### D.2  NEUROEVOLUTION EVALUATION HYPERPARAMETERS

| Brax Task Evaluation Hyperparameters | | | |
|---|---|---|---|
| Hyperparameter | Value | Hyperparameter | Value |
| Generations | 2000 | Population | 256 |
| MC Evaluations | 16 | MLP Layers | 4 |
| Hidden Units | 32 | Activation | Tanh |

The Brax experiments all used observation normalization.

| MNIST Task Evaluation Hyperparameters | | | |
|---|---|---|---|
| Hyperparameter | Value | Hyperparameter | Value |
| Generations | 4000 | Population | 128 |
| MC Evaluations | 1 | CNN Layers | 1 |
| Batchsize | 1024 | Activation | ReLU |

| MinAtar Task Evaluation Hyperparameters | | | |
|---|---|---|---|
| Hyperparameter | Value | Hyperparameter | Value |
| Generations | 5000 | Population | 256 |
| MC Evaluations | 64 | CNN Layers | 1 |
| Linear Layers | 1 | Activation | ReLU |
| Hidden Units | 32 | Episode Steps | 500 |

The MinAtar experiments do not use observation normalization.

### D.3 BASELINE SETTINGS & TUNING

Throughout the Brax control neuroevolution experiments, we consider the following baselines:

1. OpenES (Salimans et al., 2017) using Adam and centered rank fitness transformation.

2. PGPE (Sehnke et al., 2010) using Adam and centered rank fitness transformation.

3. ASEBO (Choromanski et al., 2019) using Adam and z-scored fitness transformation.

4. SNES (Wierstra et al., 2014) using the default rank-based fitness transformation.

5. sep-CMA-ES (Ros & Hansen, 2008) using an elite ratio of 0.5.

We tuned the initial search scale $\sigma_0$ for all baselines and LES by running a grid search over $\sigma_0 \in [0.05, 0.2]$. We repeat this exercise for all individual environments. Furthermore, we tuned the learning rate for OpenES (0.05), PGPE (0.02) and ASEBO (0.01) on the Brax ant environment and using a population size of 256. For OpenES we exponentially decay $\sigma$ to a minimal value of 0.01 using a decay constant of 0.999. For ASEBO the decay rate is set to $\gamma = 0.99$, $l = 150$ and for better comparability we sample full populations instead of a dynamically adjusting the population size. For PGPE the learning rate for $\sigma$ is set to 0.1 and maximal relative change of $\sigma$ is constrained to 20 percent.

For the computer vision tasks we again tune the initial scale by running a grid search over $\sigma_0 \in [0.05, 0.2]$ for each task. The learning rates are fixed as follows: PGPE (0.01), OpenES (0.01), ASEBO (0.01). Finally, for the MinAtar experiments (Figure 21) we only consider PGPE and OpenES as baselines and tune both their learning rate ($\alpha \in [0.005, 0.0075, 0.01, 0.02, 0.03, 0.04]$) and initial scale ($\sigma_0 \in [0.025, 0.05, 0.075, 0.1, 0.125, 0.15]$) by running a grid search.

Note that LES only has a single hyperparameter $\sigma_0$ and is fairly robust to its choice (see SI E.2.1).

### D.4 SOFTWARE & COMPUTE REQUIREMENTS

This project has been made possible by the usage of freely available Open Source software. This includes the following: NumPy: Harris et al. (2020), Matplotlib: Hunter (2007), Seaborn: Waskom (2021), JAX: Bradbury et al. (2018), Evosax: Lange (2022a), Gymnax: Lange (2022b), Evojax: Tang et al. (2022), Brax: Freeman et al. (2021). All experiments (both meta-training and evaluation on Brax) were implemented using JAX for parallel rollout evaluations. The simulations were run on individual NVIDIA V100 GPUs. Each MetaBBO meta-training run for 1500 meta-generations takes ca. 1 hour. The Brax evaluations require between 30 minutes (hopper, reacher, walker, fetch, ur5e) and 2 hours (ant, humanoid, halfcheetah). The computer vision evaluation experiments take ca. 10 minutes and the MinAtar experiments last for ca. 35 minutes on a V100.

# E  ADDITIONAL RESULTS

## E.1  META-EVOLUTION TRAINING CURVES & ABLATIONS

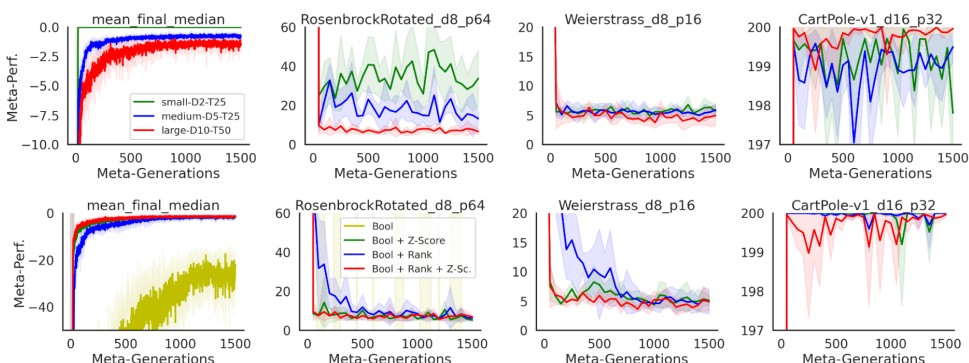

Figure 8: Comparing different meta-evolution training distributions (top) and input ablations to LES (bottom). Results are averaged over 5 independent runs (+/- 1.96 standard errors).

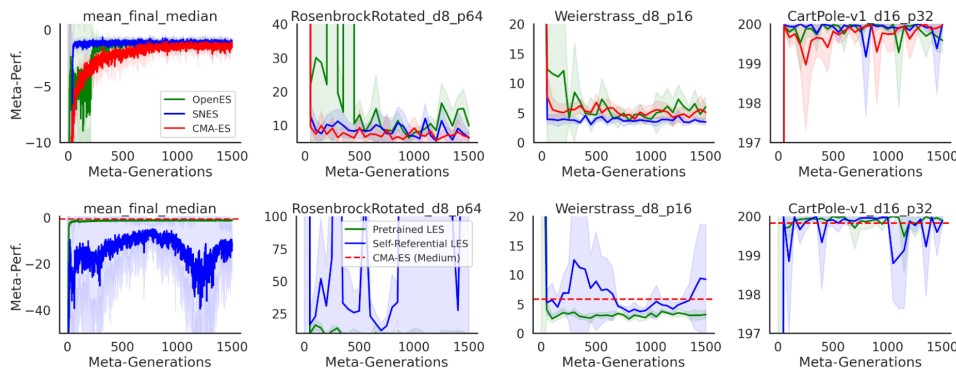

Figure 9: Comparing meta-evolution optimizers (top) and training setups (bottom, MetaBBO & self-referential MetaBBO). Results are averaged over 5 independent runs (+/- 1.96 standard errors).

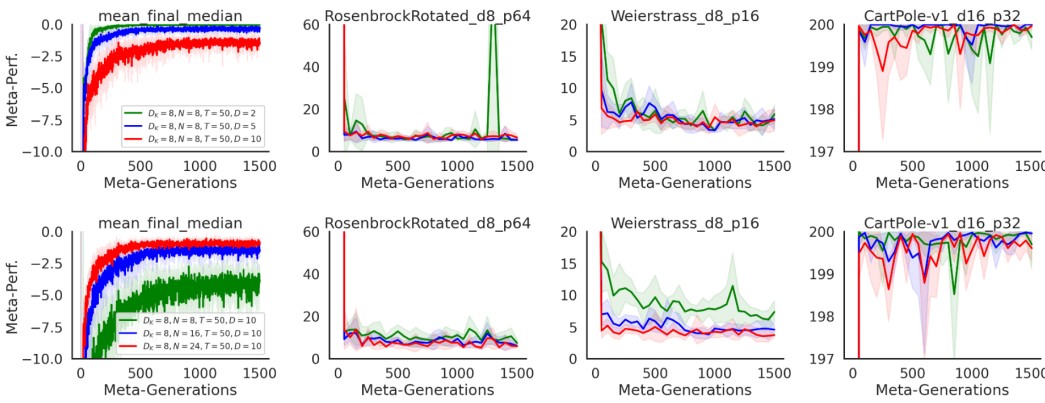

Figure 10: Comparing different meta-evolution training problem dimensions $D$ (top) and population sizes $N$ (bottom). Results are averaged over 5 independent runs (+/- 1.96 standard errors).

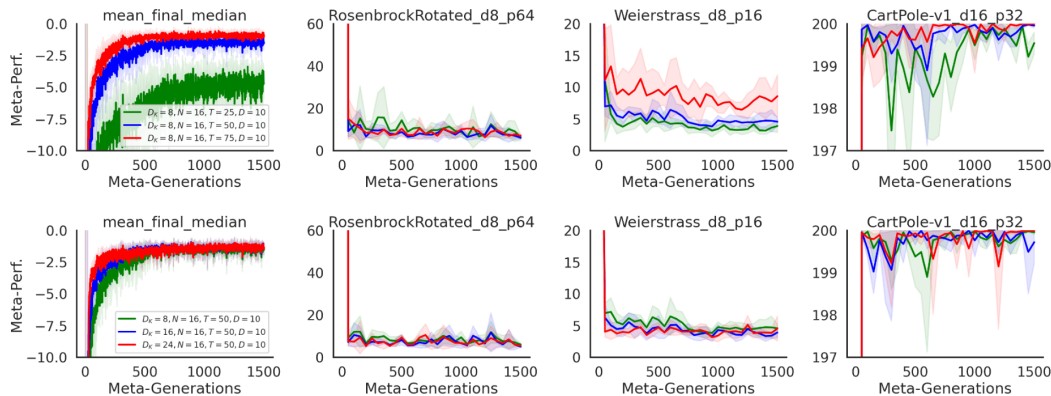

Figure 11: Comparing different meta-evolution training inner loop time horizons $T$ (top) and LES key embedding sizes $D_K$ (bottom). Results are averaged over 5 independent runs (+/- 1.96 std err.).

## E.2 DETAILED DES/LES NEUROEVOLUTION RESULTS

### E.2.1 DETAILED ABLATION/ROBUSTNESS PERFORMANCE ON BRAX TASKS

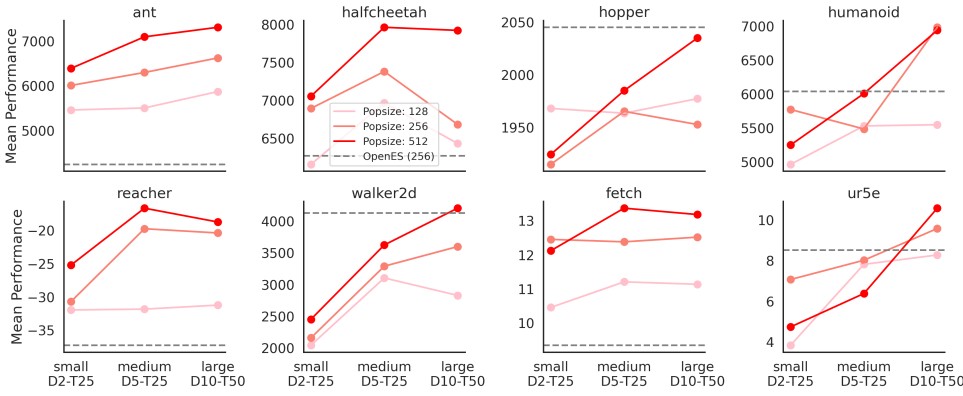

Figure 12: Impact of Meta-Task Distribution on Brax Performance. Results are averaged over 5 independent runs (+/- 1.96 standard errors).

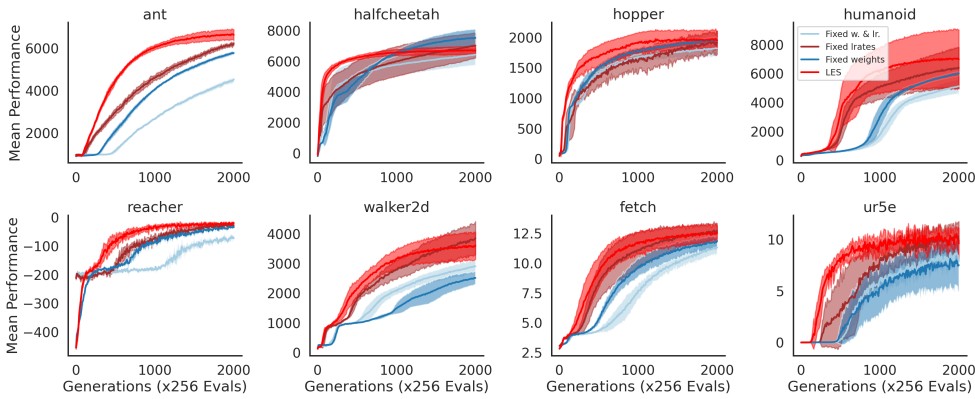

Figure 13: Brax LES Neural Network Components Ablation Study. Results are averaged over 5 independent runs (+/- 1.96 standard errors).

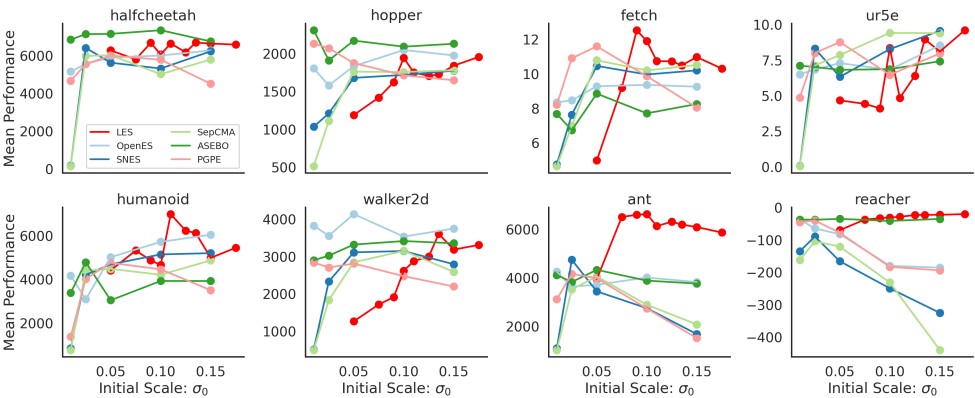

Figure 14: Robustness to initial scale of standard deviations $\sigma_0$. Results are averaged over 5 independent runs (+/- 1.96 standard errors).

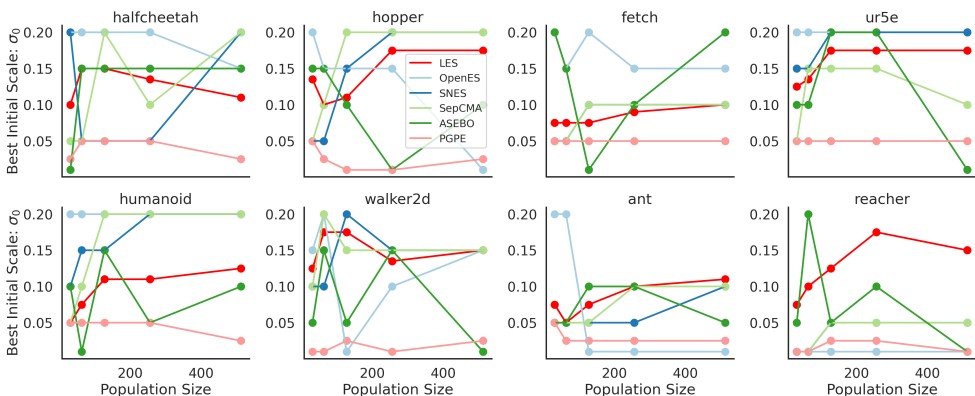

Figure 15: Best initial scale of standard deviations across population sizes. Results are averaged over 5 independent runs (+/- 1.96 standard errors).

### E.2.2 DETAILED NEUROEVOLUTION LEARNING CURVES

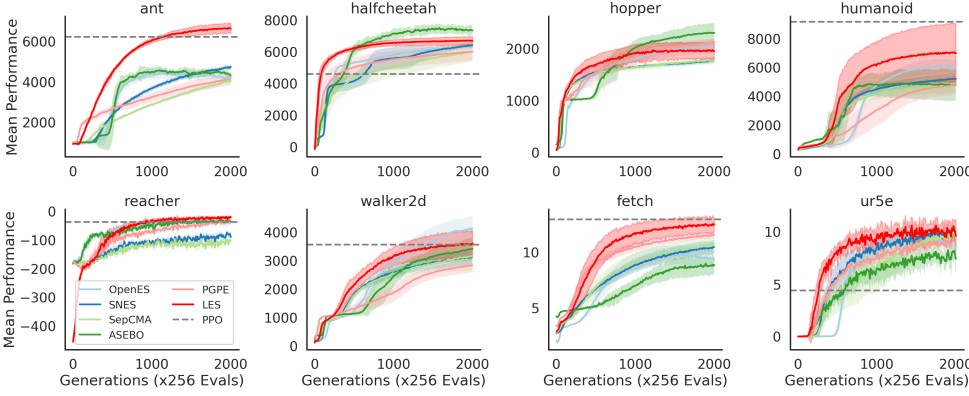

Figure 16: Detailed Brax Learning Curves with a population size of 256. PPO scores were taken from Lu et al. (2022) using default settings provided by the Brax implementation (Freeman et al., 2021). Results are averaged over 5 independent runs (+/- 1.96 standard errors).

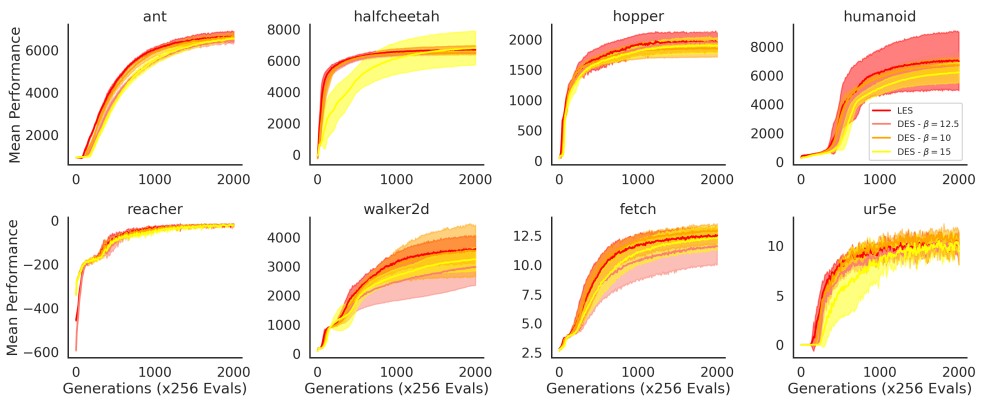

Figure 17: Detailed Brax Learning Curves with DES($\beta$) variants and a population size of 256. Results are averaged over 5 independent runs (+/- 1.96 standard errors).

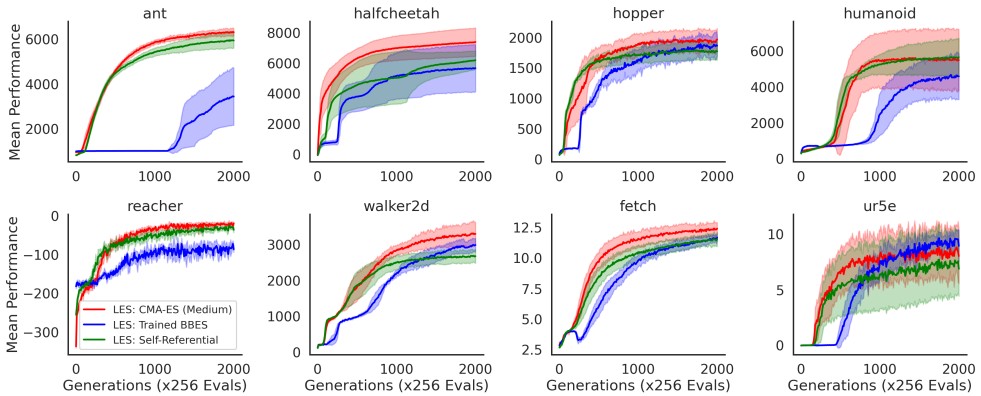

Figure 18: Detailed Brax Learning Curves with Self-Referentially Meta-Trained LES and a population size of 256. Results are averaged over 5 independent runs (+/- 1.96 standard errors).

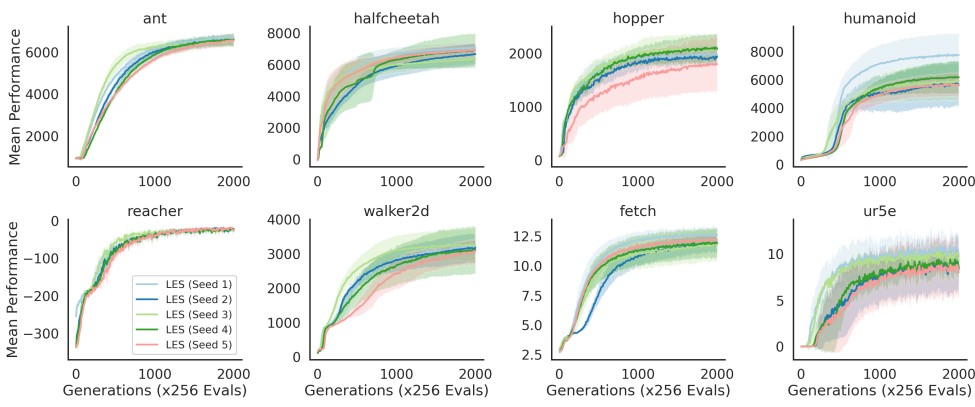

Figure 19: Detailed Brax Learning Curves with 5 separately Meta-Trained LES and a population size of 256. Results are averaged over 5 independent runs (+/- 1.96 standard errors).

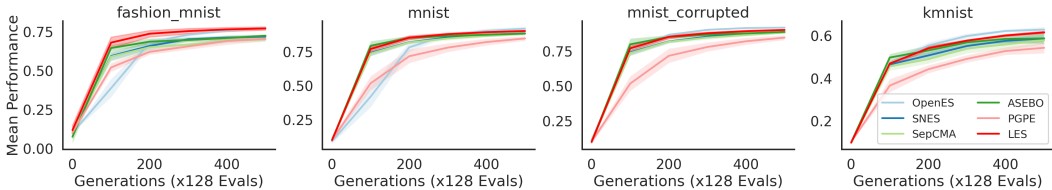

Figure 20: Detailed Learning Curves (Test Accuracy) on MNIST variants with a population size of 128. Results are averaged over 5 independent runs (+/- 1.96 standard errors).

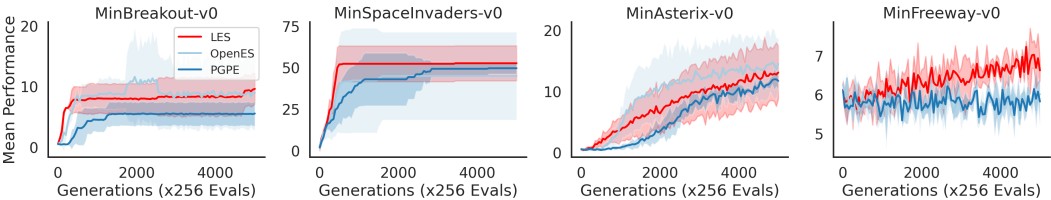

Figure 21: Detailed MinAtar Learning Curves with a population size of 256. Results are averaged over 3 independent runs (+/- 1.96 standard errors).

## F    PSEUDO-CODE IMPLEMENTATIONS FOR DES & LES

### F.1    JAX-BASED PSEUDO-CODE FOR DISCOVERED EVOLUTION STRATEGY

```python
from typing import Tuple
import chex
import jax
import jax.numpy as jnp

def get_des_weights(popsize: int, temperature: float = 12.5):
    """Compute discovered recombination weights."""
    ranks = jnp.arange(popsize)
    ranks /= ranks.size - 1
    ranks = ranks - 0.5
    sigout = nn.sigmoid(temperature * ranks)
    weights = nn.softmax(-20 * sigout)
    return weights

class DES(object):
    def __init__(
        self,
        popsize: int,
        num_dims: int,
        temperature: float = 12.5,
    ):
        """Discovered Evolution Strategy"""
        self.strategy_name = "DES"
        self.popsize = popsize
        self.num_dims = num_dims
        self.temperature = temperature

    @property
    def params_strategy(self) -> EvoParams:
        """Return default parameters of evolution strategy."""
        return EvoParams(temperature=self.temperature)
```

```python
def initialize(
    self, rng: chex.PRNGKey, params: EvoParams
) -> EvoState:
    """`initialize` the evolution strategy."""
    # Get DES discovered recombination weights.
    weights = get_des_weights(self.popsize, params.temperature)
    initialization = jax.random.uniform(
        rng,
        (self.num_dims,),
        minval=params.init_min,
        maxval=params.init_max,
    )
    state = EvoState(
        mean=initialization,
        sigma=params.sigma_init * jnp.ones(self.num_dims),
        weights=weights.reshape(-1, 1),
    )
    return state

def ask(
    self, rng: chex.PRNGKey, state: EvoState, params: EvoParams
) -> Tuple[chex.Array, EvoState]:
    """`ask` for new proposed candidates to evaluate next."""
    z = jax.random.normal(rng, (self.popsize, self.num_dims))
    x = state.mean + z * state.sigma.reshape(
        1, self.num_dims
    )
    return x, state

def tell(
    self,
    x: chex.Array,
    fitness: chex.Array,
    state: EvoState,
    params: EvoParams,
) -> EvoState:
    """`tell` update to ES state."""
    weights = state.weights
    x = x[fitness.argsort()]
    # Weighted updates
    weighted_mean = (weights * x).sum(axis=0)
    weighted_sigma = jnp.sqrt(
        (weights * (x - state.mean) ** 2).sum(axis=0) + 1e-06
    )
    mean = state.mean + params.lrate_mean * (weighted_mean - state.
mean)
    sigma = state.sigma + params.lrate_sigma * (
        weighted_sigma - state.sigma
    )
    return state.replace(mean=mean, sigma=sigma)
```

Listing 1: Pseudo-Code Discovered Evolution Strategy.

### F.2 JAX-BASED PSEUDO-CODE FOR LEARNED EVOLUTION STRATEGY

```python
from typing import Tuple
import chex
import jax
import jax.numpy as jnp
from flax import linen as nn

class AttentionWeights(nn.Module):
    """Self-attention layer for recombination weights."""

    att_hidden_dims: int = 8

    @nn.compact
    def __call__(self, X: chex.Array) -> chex.Array:
        keys = nn.Dense(self.att_hidden_dims)(X)
        queries = nn.Dense(self.att_hidden_dims)(X)
        values = nn.Dense(1)(X)
        A = nn.softmax(jnp.matmul(queries, keys.T) / jnp.sqrt(X.shape[0])
    )
        weights = nn.softmax(jnp.matmul(A, values).squeeze())
        return weights[:, None]

class EvoPathMLP(nn.Module):
    """MLP layer for learning rate modulation based on evopaths."""

    mlp_hidden_dims: int = 8

    @nn.compact
    def __call__(
        self, path_c: chex.Array, path_sigma: chex.Array, time_embed:
    chex.Array
    ) -> Tuple[chex.Array, chex.Array]:
        timestamps = jnp.repeat(jnp.expand_dims(time_embed, axis=0),
    repeats=path_c.shape[0], axis=0)
        X = jnp.concatenate([path_c, path_sigma, timestamps], axis=1)
        hidden = jax.vmap(nn.Dense(self.mlp_hidden_dims), in_axes=(0))(X)
        hidden = nn.relu(hidden)
        lrates_mean = nn.sigmoid(nn.Dense(1)(hidden)).squeeze()
        lrates_sigma = nn.sigmoid(nn.Dense(1)(hidden)).squeeze()
        return lrates_mean, lrates_sigma

class LES(object):
    """Population Invariant Black-Box Evolution Strategy."""

    def __init__(self, popsize: int, num_dims: int, les_net_params: chex.
    ArrayTree):
        self.strategy_name = "LES"
        self.popsize = popsize
        self.num_dims = num_dims
        self.evopath = EvolutionPath(num_dims=self.num_dims, timescales=
    jnp.array([0.1, 0.5, 0.9]))
        self.weight_layer = AttentionWeights(8)
        self.lrate_layer = EvoPathMLP(8)
        self.fitness_features = FitnessFeatures()
        self.les_net_params = les_net_params

    @property
    def params_strategy(self) -> EvoParams:
        """Return default parameters of evolution strategy."""
        return EvoParams(net_params=self.les_net_params)
```

```python
def initialize(self, rng: chex.PRNGKey, params: EvoParams) ->
EvoState:
    """`initialize` the evolution strategy."""
    init_mean = jax.random.uniform(
        rng,
        (self.num_dims,),
        minval=params.init_min,
        maxval=params.init_max,
    )
    init_sigma = params.sigma_init * jnp.ones(self.num_dims)
    init_path_c = self.evopath.initialize()
    init_path_sigma = self.evopath.initialize()
    return EvoState(mean=init_mean, sigma=init_sigma, path_c=
init_path_c, path_sigma=init_path_sigma)

def ask(
    self, rng: chex.PRNGKey, state: EvoState, params: EvoParams
) -> Tuple[chex.Array, EvoState]:
    """`ask` for new parameter candidates to evaluate next."""
    noise = jax.random.normal(rng, (self.popsize, self.num_dims))
    x = state.mean + noise * state.sigma.reshape(1, self.num_dims)
    x = jnp.clip(x, params.clip_min, params.clip_max)
    return x, state

def tell(
    self, x: chex.Array, fitness: chex.Array, state: EvoState, params
: EvoParams,
) -> EvoState:
    """`tell` performance data for strategy state update."""
    fit_re = self.fitness_features.apply(x, fitness, state.
best_fitness)
    time_embed = tanh_timestamp(state.gen_counter)
    weights = self.weight_layer.apply(
        params.net_params["recomb_weights"], fit_re
    )
    weight_diff = (weights * (x - state.mean)).sum(axis=0)
    weight_noise = (weights * (x - state.mean) / state.sigma).sum(
axis=0)
    path_c = self.evopath.update(state.path_c, weight_diff)
    path_sigma = self.evopath.update(state.path_sigma, weight_noise)
    lrates_mean, lrates_sigma = self.lrate_layer.apply(
        params.net_params["lrate_modulation"], path_c, path_sigma,
        time_embed
    )
    weighted_mean = (weights * x).sum(axis=0)
    weighted_sigma = jnp.sqrt(
        (weights * (x - state.mean) ** 2).sum(axis=0) + 1e-10
    )
    mean_change = lrates_mean * (weighted_mean - state.mean)
    sigma_change = lrates_sigma * (weighted_sigma - state.sigma)
    mean = state.mean + mean_change
    sigma = state.sigma + sigma_change
    mean = jnp.clip(mean, params.clip_min, params.clip_max)
    sigma = jnp.clip(sigma, 0, params.clip_max)
    return state.replace(
        mean=mean, sigma=sigma,
        path_c=path_c, path_sigma=path_sigma,
        gen_counter=state.gen_counter + 1,
    )
```

Listing 2: Pseudo-Code Learned Evolution Strategy.

