# OpenReview forum: "Discovering Evolution Strategies via Meta-Black-Box Optimization"
_ICLR.cc/2023/Conference — ICLR 2023 poster_

### Official Review · Reviewer_Pfze · 2022-10-20

**Confidence:** 4
**Correctness:** 3
**Technical Novelty And Significance:** 3
**Empirical Novelty And Significance:** 2
**Recommendation:** 8

**Clarity, Quality, Novelty And Reproducibility:**

### Clarity

The paper has been written in a relatively clear manner. However, I feel the writing can be a bit more self-contained. While the referenced publications do contain the material, I feel that briefly mentioning the core idea behind key concepts: ‘set transformers, pegasus trick, baselines, e.t.c would help a broader set of readers in grasping and critically understanding the approach

### Novelty

The work seems to be novel and the studies done through DES and other ablations provide clarity over what works and what doesn’t.

### Quality

The limitations have been briefly discussed and addressed in the conclusion as well

### Reproducibility

The compute resources on which the experiments have been run are not mentioned in the main paper or the appendix. Considering this is an ES-based strategy, I would highly recommend mentioning this.


**Strength And Weaknesses:**

### Strengths:
- The approach of using self-referential mechanisms for evolutionary optimization is interesting and seems to lend itself naturally to the ways in which evolutionary methods perform search and exploration.
- The experiments have gone in interesting directions, and have been shown n a clear and concise manner
- The literature has been covered sufficiently

### Weaknesses:
- Set transformers seem to be the differentiating factor here. It is not clear how the proposed method explicitly relates to Set Transformers, except for the invariance property being leveraged here. Is that a sufficient condition for any method that uses attention to be considered a set transformer? I would propose explicating this in a more self-contained manner
- The central question seems to be: Can an end-end method perform better than some other methods, static and dynamic, in optimizing black box functions? The comparison has been performed only on static baselines (NES replaces points with distributions, openES uses ES for control problems and RL, and sep-CMA-ES uses diagonalization). None of the baselines are dynamically tuning hyperparameters of the inner loop (something that the BB method does). An example of potential baseline in this direction could be Dynamic Algorithm Configuration (DAC) (https://ml.informatik.uni-freiburg.de/wp-content/uploads/papers/20-PPSN-LTO-CMA.pdf). In general, having a dynamic baseline would make the experimentsstronger.
- The rationale for using the LSTM for adjusting learning rates is not clearly explained, Since we see that  fixed LR ends up performing as well as the dynamic ones. Additionally, since we see that learning rates learned by an LSTM are not having a significant impact when compared with fixed ones, what I am not sure of is whether a learning rate schedule policy would perform better or not (https://arxiv.org/pdf/2007.04223.pdf).
- The selection of hyperparameters : T=50, N=16, D, z , e.t.c - is not explained. How did the authors decide these to be the relevant ones, and is the method sensitive to these values? Considering the argument for an end-end method, I think this question becomes more relevant


**Summary Of The Paper:**

The paper introduces a method for Evolutionary Search (ES) that works by sampling hyperparameters for Learned Evolutionary strategies (LES) over different tasks and then uses self-attention to compute the recombination weights. The authors compare this approach to Handcrafted ES baselines (openES, SNES and sep-CMA-ES), and ablate their method to study the impact of the different components of Neural Networks. Based on this ablation, they derive a heuristic which is incorporated into an algorithm called Discovered Evolutionary Strategy (DES). Finally, the authors bootstrap LES using LES (argued as self-reference).

**Summary Of The Review:**

The idea is novel and interesting and I definitely see the relevance to the meta-learning community. Evolutionary strategies have always been an interesting direction to gradient-based optimization, and so, I feel the research direction of end-end meta-optimization in a self-referential manner is interesting, to say the least. However, I feel the work is not yet ready for publication. This is primarily due to missing stronger baselines. I think the rationale for using a self-referential loop i.e. why should we use implicit models to begin with needs to be sufficiently justified with comparisons against static and dynamic baselines and that is currently lacking from the work.

---

> ### Author Response · Authors · 2022-11-13
> **Reply - Reviewer Pfze**
>
> We thank the reviewer for their detailed suggestions, appreciation of the detailed analysis and critical assessment.
>
> “[...] It is not clear how the proposed method explicitly relates to Set Transformers, except for the invariance property being leveraged here.”
>
> The proposed LES architecture indeed only makes use of the equivariance implied by the dot-product self-attention. Nonetheless, our original motivation derived from viewing the ES update equation as a set operation, which requires invariance to the population order. It turned out that we did not need all of the Transformer components (layer norm, dropout, large MLP post-processing) to meta-learn a SotA ES for neuroevolution. We state that the LES architecture can be viewed as a “minimal Set Transformer” due to the historical development of the method and because we think that the neuroevolution community can benefit from this new ES parametrization perspective.
>
> “[...] The comparison has been performed only on static baselines (NES replaces points with distributions, openES uses ES for control problems and RL, and sep-CMA-ES uses diagonalization).”
>
> The current manuscript has been extended to include two more competitive baselines (Figure 1, 3, 14): PGPE (Sehnke et al., 2010) and ASEBO (Choromanski et al., 2019). ASEBO adapts the sampling procedure to the fitness geometry by iteratively estimating gradient subspaces. The method can thereby be viewed as dynamic.
>
> The DAC approach by Shala et al. is definitely related work, but the direct comparison may not be applicable since the meta-training involves additional Guided Policy Search using a teacher ES (e.g. CMA-ES). The DAC method also assumes a fixed population size (in the paper 10), which makes comparisons harder. Our trained LES can be applied to any population size of problem dimensionality without re-training. Part of our main contribution is the strong generalization of a single meta-trained ES across population sizes, number of dimensions and train horizons/number of generations.
>
> “[...] The rationale for using the LSTM for adjusting learning rates is not clearly explained, Since we see that fixed LR ends up performing as well as the dynamic ones. Additionally, since we see that learning rates learned by an LSTM are not having a significant impact when compared with fixed ones, what I am not sure of is whether a learning rate schedule policy would perform better or not.”
>
> We were wondering the same and observed a form of time horizon overfitting in the modulation (especially for the ur5e control task). During this rebuttal period we therefore made a couple of modifications to the learning rate modulation: We replaced the LSTM learning rate modulation with a MLP and additionally sampled a start time t during meta-training (before we always started an inner loop with t=0).
>
> After doing so, we see a bigger positive effect of the learning modulation (see updated Figure 5, ablation). That being said, the modulation does not yield a clear schedule as commonly known from gradient-based optimization. Instead, it rather infers a single optimal value and dampens an initial overshooting of the Gaussian scale parameters.
>
> “[...] The selection of hyperparameters : T=50, N=16, D, z , e.t.c - is not explained. How did the authors decide these to be the relevant ones, and is the method sensitive to these values? Considering the argument for an end-end method, I think this question becomes more relevant”
>
> We agree and apologize for the lack of clarification. We added additional meta-training studies in the SI (Figures 20 and 21), which compares different inner loop time horizons (T), population sizes (N), problem dimensions (D)  and self-attention keys dimensions (D_K). The meta-learning process is largely robust to these choices with a few notable exceptions (see performance on Weierstrass task with D=8 and N=16): Too few population members (N=8) can hurt performance and too long inner loops (T=75) can do so as well. Interestingly, increasing the model size does not improve performance. We therefore chose the smallest model to reduce the memory footprint.

---

> > ### Comment · Reviewer_Pfze · 2022-11-24
> > **Increasing my rating**
> >
> > Dear authors,
> >
> > Thank you very much for your reply and improved revision.
> > My doubts have been resolved by your efforts and thus, I increased my voting to "accept".

---

> > > ### Author Response · Authors · 2022-12-05
> > > **Thank you!**
> > >
> > > Dear reviewer Pfze,
> > >
> > > Thank your reply. We are glad that our revised work could address your concerns.
> > >
> > > Have a good week,
> > >
> > > The authors.

---

### Official Review · Reviewer_RKfF · 2022-10-23

**Confidence:** 4
**Correctness:** 4
**Technical Novelty And Significance:** 2
**Empirical Novelty And Significance:** 2
**Recommendation:** 6

**Clarity, Quality, Novelty And Reproducibility:**

This is a well-organized paper with high quality. However, as the general idea of this paper is using learning techniques to replace the heuristic components in optimization methods, which is not new, the novelty of this paper is limited. The experiments in this work are well-motivated with good introductions, figures and explanations, but not convincing (please see my comment in the above “weakness” part). The reproducibility looks good to me.

**Strength And Weaknesses:**

Strength:

This paper is well-written and easy to follow. Experiments show the effectiveness of the proposed method from various views.

Weakness:

1. Even that the learning of update rules in ES has not been studied in a targeted way, the general idea of using learning techniques to replace the heuristic components in optimization methods is not new [1], which limits the novelty of this paper. I did not find much new insight from this work. There have been many works to learn hyper-parameters of some complex optimization methods.

2. The authors should clearly explain and show the advantage of the proposed method (i.e., using a transformer to represent the update rule of ES and applying again BBO (e.g., CMA-ES) to optimize the parameters of the transformer) over other general ways of learning the heuristic rules. Can you show the benefit of using transformer empirically? How about the cost of the proposed method, i.e., training the transformer? As mentioned in the paper, the self-attention mechanism scales quadratically in the number of population members (in section 8). Meanwhile, it seems that a relatively large population size is important (in figure 4), which may limit the application of the proposed method.

3. More advanced ES methods besides OpenES and SepCMA like [2-4] are expected for comparison to show that the learning of the update rules can obtain a competitive performance w.r.t. SOTA ES methods.

[1] Marcin Andrychowicz et al, Learning to Learn by Gradient Descent by Gradient Descent, NIPS’16.
[2] Guided evolutionary strategies: Augmenting random search with surrogate gradients. ICML’19
[3] From complexity to simplicity: Adaptive ES-active subspaces for blackbox optimization. NeurIPS’19
[4] Self-guided evolution strategy with historical estimated gradients, IJCAI’20.


**Summary Of The Paper:**

The update rules of Black-Box Optimization (BBO) methods are traditionally formalized by hand-craft heuristics, which is inflexible. To tackle this problem, this paper proposes to parameterize the update rules of a typical BBO algorithm, Evolution Strategy (ES), by a transformer, and use meta-learning together with BBO techniques to optimize it, obtaining the Learned Evolution Strategy (LES). Experiments on BBOB benchmark, classification on MNIST and continuous control tasks generally show the effectiveness of the proposed framework. Furthermore, this paper shows that the proposed LES can be optimized through a self-referential style.

**Summary Of The Review:**

This paper proposes a new method to learn the update rules of ES, with relatively various experiments to show its effectiveness. However, the general idea of this paper, using learning techniques to replace the heuristic components in optimization methods, is not new, but another standard implementation (here the authors use transformer to represent the rules and optimize the parameters of transformer by CMA-ES). As listed in the weakness, the lack of comparison with advanced ES methods and the scalability issues also limit the contribution of this paper.

---

> ### Author Response · Authors · 2022-11-13
> **Reply - Reviewer RKfF**
>
> We thank the reviewer for their detailed suggestions, appreciation of the manuscript writing and improvement recommendations.
>
> “[...] Even that the learning of update rules in ES has not been studied in a targeted way, the general idea of using learning techniques to replace the heuristic components in optimization methods is not new [1], which limits the novelty of this paper. I did not find much new insight from this work. There have been many works to learn hyper-parameters of some complex optimization methods.”
>
> To be fair – the same argument could in principle be applied to any meta-learning paper, which aims to discover inductive biases using bi-level optimization. Our main contributions do not lie in the meta-learning procedure itself, but in the introduction of a novel learnable ES parameterization using dot-product self-attention over population members. Furthermore, our meta-learned strategy outperforms very competitive baseline ES on many challenging neuroevolution tasks. A surprising insight to us was the strong generalization properties of LES: Meta-training on low-dimensional BBOB problems for short time-horizons and with small populations yields a SotA neuroevolution method. We also reverse engineered the meta-learned ES into a closed-form DES. To the best of our knowledge, the discovered update rule was previously unknown and thereby provides a contribution in itself. Finally, the successful self-referential meta-training procedure of ES was previously not established.
>
> “[...] Can you show the benefit of using transformer empirically? How about the cost of the proposed method, i.e., training the transformer?”
>
> In the ablation study (Figure 5 top left and SI Figure 11) we ablate the Transformer-based recombination weight module and find that it provides most of the performance gains observed (as compared to the learning rate modulation). Furthermore, we outperform non-Transformer-based baseline ES.
>
> We are sorry to have missed a paragraph on the computational costs of our meta-training procedure and have added a new section in SI (A.9). The meta-training is done on a single V100 GPU and does not take long (ca. 1 hour). The vast majority of computational resources were spent on the post-meta-training evaluation. We believe that this points towards the strong and effective inductive bias provided by the self-attention-based parameterization.
>
> “[...] As mentioned in the paper, the self-attention mechanism scales quadratically in the number of population members (in section 8). Meanwhile, it seems that a relatively large population size is important (in figure 4), which may limit the application of the proposed method.”
>
> The quadratic scaling of self-attention does not pose a practical limitation for common population sizes (e.g. <10k) and can easily be performed even on CPU devices. Note that the self-attention computation boils down to three tall matrix multiplications and a row-wise non-linearity. The majority of FLOPs is spent on the population member evaluations, which are independent of the ES update rule. Furthermore, all advances in scalable Transformers can in principle be applied (e.g. Wang et al. (2020), Jaegle et al. (2021)).
>
> “[...] More advanced ES methods besides OpenES and SepCMA like [2-4] are expected for comparison to show that the learning of the update rules can obtain a competitive performance w.r.t. SOTA ES methods. [...] As listed in the weakness, the lack of comparison with advanced ES methods and the scalability issues also limit the contribution of this paper.”
>
> In order to address the reviewer's concerns, we have added the proposed ASEBO (Choromanski et al., 2019) baseline and additional PGPE baseline (Sehnke et al., 2010). The meta-trained LES still outperforms all baselines. Guided ES (Maheswaranathan et al., 2019) requires a surrogate gradient, which is not directly available for the considered neuroevolution tasks. We attempted using the finite-difference gradient from antithetic sampling, but could not obtain competitive results. Therefore, we chose to omit these additional results, since they are not directly comparable to LES or the other baselines.
>
> References:
>
> Wang, Sinong, et al. "Linformer: Self-attention with linear complexity." arXiv preprint arXiv:2006.04768 (2020).
>
> Jaegle, Andrew, et al. "Perceiver: General perception with iterative attention." International conference on machine learning. PMLR, 2021.
>
> Sehnke, Frank, et al. "Parameter-exploring policy gradients." Neural Networks 23.4 (2010): 551-559.
>
> Choromanski, Krzysztof M., et al. "From complexity to simplicity: Adaptive es-active subspaces for blackbox optimization." Advances in Neural Information Processing Systems 32 (2019).
>
> Maheswaranathan, Niru, et al. "Guided evolutionary strategies: Augmenting random search with surrogate gradients." International Conference on Machine Learning. PMLR, 2019.

---

> > ### Comment · Reviewer_RKfF · 2022-12-06
> > **Thanks for your reply**
> >
> > Thanks for your reply. Some of my concerns have been addressed, but the empirical comparison is still not strong enough in my opinion. I believe that Guided ES (Maheswaranathan et al., 2019) and Self-guided ES (Liu et al., 2020) should be compared, which are representative ES methods (which estimate the gradient by sampling) for BBO. Furthermore, as stated in my comments, the general idea of the method is not very novel, and I did not find much impact and new insight. Thus, I will keep my score.

---

> > > ### Author Response · Authors · 2022-12-06
> > > **Reply - Reviewer RKfF (contd.)**
> > >
> > > We thank the review for their response and acknowledgement of our additional work. We are also glad to hear that we could resolve some of their concerns, but believe that there may remain a misunderstanding: As requested by the reviewer, we added the ASEBO (Choromanski et al., 2019) baseline for the neuroevolution tasks. ASEBO estimates a subspace based on the history of finite-difference gradient updates. Our proposed meta-learned LES beats ASEBO on the majority of tasks. Furthermore, we added an additional PGPE baseline (Sehnke et al., 2010) and conducted several additional ablations (meta-task distribution and self-attention architecture) as well as experiments on visual RL tasks (4 MinAtar environments). All our qualitative results are strengthened by these additional experiments.
> > >
> > > Guided ES (Maheswaranathan et al., 2019) is similar in spirit, but does rely on the availability of a surrogate gradient (e.g. noisy version of the true gradient with respect to the search distribution mean). A direct comparison is therefore not applicable and has to the best of our knowledge only been done in Liu et al. (2020). They use the finite-difference gradient as a surrogate, which we also attempted but found that this did not result in a competitive baseline. This is in line with their experiments, in which Guided ES performs worse than OpenAI-ES ("Vanilla ES", see page 1479 Figure 3). Our proposed LES though outperforms OpenAI-ES. We therefore decided to omit the Guided ES results.
> > >
> > > We believe that our work is novel and can guide the future directions of meta-learning ES: We are the first to propose a self-attention-based parametrization of ES families. Furthermore, we conduct extensive experiments on more than 15 benchmark tasks, which show that the proposed meta-learned LES can outperform 5 competitive baseline ES. These baselines include estimation-of-distribution methods (Sep-CMA-ES), natural gradient methods (SNES) and finite-difference gradient-based methods (OpenAI-ES, PGPE), which also leverage gradient information for sampling (ASEBO).

---

> > > > ### Comment · Reviewer_RKfF · 2022-12-06
> > > > **Thanks for your further clarification**
> > > >
> > > > Thanks for your further clarification. I suggest you include Self-guided ES (Liu et al., 2020) into the comparison. I have increased my score accordingly.

---

> ### Author Response · Authors · 2022-12-05
> **Request for rebuttal assessment/feedback**
>
> Dear reviewer RKfF,
>
> We again thank you for your time and feedback. Given that the ICLR discussion period will come to an end on December 12th, we kindly want to ask whether you could assess our rebuttal? We want to highlight the extensive revision of the paper (e.g. clarifications in the related work section) and the accompanying additional experiments (2 more baselines, additional CNN neuroevolution experiments on MinAtar tasks and several ablations/meta-training comparisons in the SI).
>
> We again thank the author for their constructive criticism and believe to have commented on/addressed the 3 points listed as weaknesses.
>
> Please let us know if any further questions may have come up.
>
> Have a good week,
>
> The authors.

---

### Official Review · Reviewer_cHGq · 2022-10-25

**Confidence:** 3
**Clarity, Quality, Novelty And Reproducibility:** Please see above.
**Correctness:** 3
**Technical Novelty And Significance:** 2
**Empirical Novelty And Significance:** 3
**Recommendation:** 6

**Strength And Weaknesses:**

Strengths:
The paper is easy to follow, which is excellent for a meta-training paper, with limited jargon and many implementation details. The strengths of the paper don’t lie in the idea itself but rather in the study of its extensions and limitations. This is a well-studied idea, and the experiments illustrate well the points the authors try to make, should it be about interpreting the LES or self-referencing of the LES. In particular, the study of the meta-training task distribution is very insightful and has not been done in many meta-learning papers. Noticeably, the authors can interpret the LES to devise a Discovered ES that works in some of their experiments.

Weaknesses:

Some weaknesses are outlined by the paper itself, such as scaling the self-attention mechanism or that this algorithm works only for diagonal Gaussian ES.  It is worth pointing out also that the features used in the self-attention mechanism F_t are manually created, i.e. it is a choice made in the algorithm that the self-attention would consider these only three statistics derived from fitness. This works in practice, as shown in experiments but looks like a significant limitation in terms of generalization. The authors mention this as a limit to their approach, but the manual engineering of these features is another.

Questions:
- Would it be possible to replace the manually engineered features F_t with a learned feature map?
- Would it be possible to relax the diagonal assumption of the Gaussian covariance matrix but use, for instance, a sparsity regulator to keep it maybe sparser than full?
- In the experiments, the authors show that the method can adapt to scenarios unseen in the meta-training task distribution and generalize to higher dimensions and different loss surfaces. However, there are still, of course, tasks in which it performs poorly. Of course, one cannot expect a meta-model to perform perfectly on all tasks, and any such pretrained model will suffer from distributional shift, and these cases call for a fine-tuning procedure. How could this approach be fine-tuned, and how long would it take to adapt it for instance, to a task on which it performs poorly straight out of meta-training?
- How do you explain the difference in Brax Performance in Figure 3 bottom middle and right between the medium budget and large budget on tasks like hopper and ur5e or walker2d? Otherwise, the experiments are well-designed and conclusive. The ablation studies are particularly insightful.

**Summary Of The Paper:**

The authors introduce a method to learn the parametrization of an Evolution Strategy through the meta-training of a self-attention-based architecture. They demonstrate the generalization of their meta-training procedure to test tasks that are different from the meta-training tasks. They further analyze the limitations of their proposed algorithm and deduce an interpretable parametrization of the ES.

The approach uses a self-attention mechanism to learn the parameters of an ES as well as an LSTM architecture to learn the learning rate of the ES parameter updates from one generation to another. Their approach is by construction invariant to reordering. It also allows them to meta-train on multiple dimensions, which can then be leveraged at test time.

**Summary Of The Review:**

Please see above.

---

> ### Author Response · Authors · 2022-11-13
> **Reply - Reviewer cHGq**
>
> We thank the reviewer for their thoughtful comments and appreciation. We would like to take the opportunity to discuss their suggestions and questions.
>
> > “[...] It is worth pointing out also that the features used in the self-attention mechanism F_t are manually created, i.e. it is a choice made in the algorithm that the self-attention would consider these only three statistics derived from fitness. [...] Would it be possible to replace the manually engineered features F_t with a learned feature map?””
>
> The chosen feature construction ensures that the LES is capable of generalization across different fitness scales. In principle, one could try to use a floating point number tokenization as recently proposed by d'Ascoli et al. (2022). In this case the Transformer would have to implicitly learn to rank fitness score tokens. Another alternative could be to use random projections of the raw fitness and to use layer normalization before feeding the transformation into the self-attention layer.
>
> Finally, we believe that the explicit feature construction also enables design flexibility: One could imagine adding additional features about agent trajectories to obtain a learned quality-diversity algorithm. We are very much looking forward to investigating these extensions in future work.
>
> In Figure 8 (second row) we further investigate an ablation of the different features and show that LES is robust and can perform BBO using only a single measure of relative performance within a population. Finally, all popular ES perform fitness shaping before calculating search updates.
>
> > “[...] Would it be possible to relax the diagonal assumption of the Gaussian covariance matrix but use, for instance, a sparsity regulator to keep it maybe sparser than full?”
>
> Great question! In principle, the recombination weight network can also be plugged into covariance matrix adaptation methods. We have not done so ourselves, but believe that this is fruitful future work. Dynamically adjusting the sparsity or number of ranks in the covariance matrix sounds promising, but also potentially hard to meta-learn in a way that generalizes across numbers of problem dimensions. Finally, oftentimes covariance estimation can be challenging for neuroevolution tasks and diagonal methods outperform full covariance methods.
>
> > “[...] How could this approach be fine-tuned, and how long would it take to adapt it for instance, to a task on which it performs poorly straight out of meta-training?”
>
> The initial scale $\sigma_0$ of the LES can be adapted for the specific problem. Furthermore, we believe that there is potential for ensembling different meta-trained LES. With this rebuttal we added a new SI Figure 17, which compares meta-trained LES resulting from different random seeds. While all of them perform well, there appears to be a degree of specialization. It may be possible to meta-train different expert LES (one for multi-modal problems, one for highly skewed fitness landscapes, etc.) and to learn a hierarchical controller over them. Finally, for the ‘white-box’ DES the temperature can be tuned to control the degree of truncation.
>
> “[...] How do you explain the difference in Brax Performance in Figure 3 bottom middle and right between the medium budget and large budget on tasks like hopper and ur5e or walker2d? Otherwise, the experiments are well-designed and conclusive. The ablation studies are particularly insightful.
>
> We thank the reviewer for their positive feedback. After meta-training a LES with MLP learning rate modulation (see general comments), we no longer observe these budget “inconsistencies”. At this point we believe that they may have been due to two reasons: Time horizon overfitting of the learning rate modulation and/or the fact that the finite-difference ES baselines tend to scale well with larger population sizes.
>
> References:
>
> d’Ascoli, Stéphane, et al. "Deep symbolic regression for recurrence prediction." International Conference on Machine Learning. PMLR, 2022.

---

> ### Author Response · Authors · 2022-12-05
> **Request for rebuttal assessment/feedback**
>
> Dear reviewer cHGq,
> We again thank you for your time and feedback. Given that the ICLR discussion period will come to an end on December 12th, we kindly want to ask whether you could assess our rebuttal? We want to highlight the extensive revision of the paper (e.g. clarifications in the related work section) and the accompanying additional experiments (2 more baselines, additional CNN neuroevolution experiments on MinAtar tasks and several ablations/meta-training comparisons in the SI). We again thank the author for their stimulating questions and hope to have addressed these.
>
> Please let us know if any further questions may have come up.
> Have a good week,
> The authors.

---

> > ### Comment · Reviewer_cHGq · 2022-12-05
> > **Thank you**
> >
> > I thank the authors for the rebuttal. It has addressed some of my concerns. The addition of the baseline is very useful. I will keep my score as is.

---

### Official Review · Reviewer_Fevx · 2022-11-01

**Confidence:** 4
**Correctness:** 2
**Technical Novelty And Significance:** 2
**Empirical Novelty And Significance:** 2
**Recommendation:** 6

**Clarity, Quality, Novelty And Reproducibility:**

The paper is more or less clearly written.

The technical quality and originality (including absence of theoretical treatments) are not up to the expected level of a conference like ICLR.

**Strength And Weaknesses:**

Strength: The paper addresses a timely and ambitious research topic.

Weakness:

Statements like "We show that metaevolving this system on a small set of representative low-dimensional analytic optimization problems is sufficient to discover new evolution strategies capable of generalizing to unseen optimization problems, population sizes and optimization horizons"  - are very strong and have not been at all supported through any theoretical treatment. The generalization of LES to unseen neuroevolution tasks even in disparately high dimensions is not explained via theory of insightful empirical evidences.

Related works ignored:

Vishnu TV, Pankaj Malhotra, Jyoti Narwariya, Lovekesh Vig, and Gautam Shroff. 2019. Meta-Learning for Black-Box Optimization. In Machine Learning and Knowledge Discovery in Databases: European Conference, ECML PKDD 2019, Würzburg, Germany, September 16–20, 2019, Proceedings, Part II. Springer-Verlag, Berlin, Heidelberg, 366–381. https://doi.org/10.1007/978-3-030-46147-8_22

Choice of the functions in table A.1 for training is not adequately justified by analysis of the funcional landscape features.

It is not clear how the method achieves diversity in meta-training task distributions, no theoretical treatment was included.

The results were not compared against the best known algorithms in the pertinent areas from other genre.

Why were the results not validated through appropriate non-parametric statistical tests?

**Summary Of The Paper:**

The paper presents a self-attention based meta learning framework for the learned Gaussian ESs and tests the performance on neuroevolution tasks. The metaBBO module includes steps like Meta-sampling, Inner loop search, meta normalize, and Meta updating. Limited evaluation results on standard datasets were provided.


**Summary Of The Review:**

A meta learning framework is extended for Gaussian ESs without very exhaustive set of experiments and any theoretical results. In my view, the paper does not meet the high standards expected for ICLR.

---

> ### Author Response · Authors · 2022-11-13
> **Reply - Reviewer Fevx**
>
> We thank the reviewer for their time, feedback and questions.
>
> > “[...] A meta learning framework is extended for Gaussian ESs without very exhaustive set of experiments and any theoretical results.”
>
> We exhaustively validate all of our claims on many different problem classes (BBOB, continuous control, computer vision), tasks, population sizes, number of parameters to optimize and time horizons (e.g. Figure 1, 2, 3). To be more precise, we consider more than 15 tasks, a large range of population sizes (N in 16 to 512), problem dimensions (D in 2 to 5k), time horizons (see learning curves in SI) and multiple random seeds (>5 runs independent runs). Our meta-trained LES robustly outperforms competitive tuned baselines across the vast majority of settings.
>
> To further address the reviewer's concern, we substantially enhanced our analysis: We have added an additional set of visual control evaluation tasks (see SI Figure 19) and two more baselines (PGPE; Sehnke et al., 2010 and ASEBO; Choromanski et al., 2019). LES outperforms all considered baselines and can robustly evolve CNN policy weights on MinAtar (Young & Tian, 2019).
>
> > “[...] Statements like "We show that metaevolving this system on a small set of representative low-dimensional analytic optimization problems is sufficient to discover new evolution strategies capable of generalizing to unseen optimization problems, population sizes and optimization horizons" - are very strong and have not been at all supported through any theoretical treatment.”
>
> The ES literature is generally lacking strong theoretical results. E.g. to the best of our knowledge there is no informative convergence proof for ES like CMA-ES (e.g. as stated in Hansen et al., 2015). Furthermore, theoretical results are non-trivial to obtain for the LES, which uses a non-linear attention layer to compute recombination weights. In future work we hope to be able to provide more theoretical insights for the reverse-engineered DES without neural network components.
>
> > “[...] Vishnu TV, Pankaj Malhotra, Jyoti Narwariya, Lovekesh Vig, and Gautam Shroff. 2019. Meta-Learning for Black-Box Optimization. In Machine Learning and Knowledge Discovery in Databases: European Conference, ECML PKDD 2019”
>
> We thank the reviewer for highlighting this reference, which we missed. It is now added to our updated related work section. TV et al. meta-train RNN-based black-box optimizers, which can only perform BBO on problems with dimensionality corresponding to their meta-training setting. Furthermore, they only evaluate their system on 2 and 6 dimensional problems. LES, on the other hand, is the first meta-learned ES to successfully scale to >5k dimensions and beats established BBO methods on challenging neuroevolution tasks.
>
> > “[...] Choice of the functions in table A.1 for training is not adequately justified by analysis of the funcional landscape features. It is not clear how the method achieves diversity in meta-training task distributions, no theoretical treatment was included.”
>
> In Table 1 we list the properties of the different BBOB meta-training functions. They were carefully chosen to cover separable, moderate/high condition number and multi-modal optimization problems. Furthermore, these functions have been used for a long time in order to assess the performance of ES. In Figure 4 we show that these diverse meta-training properties help meta-learn better LES. With this revision we added a visualization of the fitness landscapes for 5 of the meta-training BBOB functions. We hope that this addresses the reviewer’s concerns.
>
> > “[...] The results were not compared against the best known algorithms in the pertinent areas from other genre.
> Why were the results not validated through appropriate non-parametric statistical tests?”
>
> We benchmark against 5 established ES including OpenES, PGPE, sep-CMA-ES, SNES and ASEBO. These are to the best of our knowledge the best known ES baselines. For each experiment we provide at least 5 independent runs and plot both mean and standard confidence intervals (1.96 standard errors). t-test decisions for statistical significance can therefore be extracted directly from the plots. All of the detailed learning curves can be viewed in detail in the SI. Can you clarify what is meant by non-parametric tests?
>
> References:
>
> Sehnke, Frank, et al. "Parameter-exploring policy gradients." Neural Networks 23.4 (2010): 551-559.
>
> Choromanski, Krzysztof M., et al. "From complexity to simplicity: Adaptive es-active subspaces for blackbox optimization." Advances in Neural Information Processing Systems 32 (2019).
>
> Young, Kenny, and Tian Tian. "Minatar: An atari-inspired testbed for thorough and reproducible reinforcement learning experiments." arXiv preprint arXiv:1903.03176 (2019).
>
> Hansen, Nikolaus, Dirk V. Arnold, and Anne Auger. "Evolution strategies." Springer handbook of computational intelligence. Springer, Berlin, Heidelberg, 2015. 871-898.

---

> > ### Comment · Reviewer_Fevx · 2022-12-13
> > **Response to Authors' Feedback**
> >
> > Dear Authors, Thanks for all your efforts to improve the paper. Most of my concerns have been resolved. However, I am a bit surprised that you are unfamiliar with non-parametric statistical hypothesis testing like the famous Wilcoxon's signed rank test to assess the statistical significance of your comparative results while comparing several stochastic optimisation algorithms. Such tests, as opposed to parametric tests like the t-test, do not assume the shape of the underlying probability distribution. You may see the following link:
> > https://www.sciencedirect.com/science/article/pii/S2210650219302639
> >
> >  Given the revision efforts by the authors and the scaling performance of the LES, I would increase my score to 6. However, I still need to be convinced enough about the novelty of the proposal and its theoretical significance, given a conference like ICLR.

---

> > > ### Author Response · Authors · 2022-12-13
> > > **Reply II - Reviewer Fevx**
> > >
> > > Dear reviewer Fevx,
> > >
> > > We thank the reviewer for their reply and acknowledgment of our substantial rebuttal revisions.
> > >
> > > > “Such tests, as opposed to parametric tests like the t-test, do not assume the shape of the underlying probability distribution.”
> > >
> > > We deeply share the opinion that rigorous statistical testing is important for progressing research and thank the reviewer for bringing the mentioned reference to our attention. All our main neuroevolution control experiments (e.g. Figure 1, 4) used 10 random seeds. This includes all baselines and is substantially more than what is commonly done for RL tasks (ca. <5 runs; see Agarwal et al., 2021 Figure 1).
> > >
> > > Our decision to plot confidence intervals with 1.96 standard errors was motivated by allowing the reader to validate the significance of our method across time/number of generations. The proposed rank-based testing would have been not easily visualized in combination with the learning curve plots. We further provide all detailed curves in the supplementary information (see Figure 14), so that the reader can investigate the algorithm ranks across environments and time: Our proposed LES performs best (rank 1) on 4 out of 8 tasks (ant, humanoid, reacher, fetch) and never consistently drops below rank two. We hope that this explains and justifies our choice.
> > >
> > > > Given the revision efforts by the authors and the scaling performance of the LES, I would increase my score to 6. However, I still need to be convinced enough about the novelty of the proposal and its theoretical significance, given a conference like ICLR.
> > >
> > > Our work is novel and significant in several aspects: We are the first to show that meta-learned ES can achieve SotA performance on challenging high-dimensional neuroevolution tasks. All previous works (e.g. Shala et al., 2020; TV et al., 2019) were limited to fixed population sizes or problem dimensionalities. Our proposed self-attention-based parametrization of the ES update rule, on the other hand, can naturally be applied to different problem settings and budgets. We believe that our work highlights the strong potential of viewing the ES update as a learnable set operation, which can be parametrized by self-attention.
> > >
> > > Furthermore, we show that classic BBO problems provide a good meta-training distribution, which enables transfer to unseen neural network tasks. This type of generalization of meta-learned BBO was previously not investigated. We provide rigorous empirical evidence that LES performs strongly on unseen BBOB functions and more than 15 neuroevolution tasks with different networks (8 control, 4 classification & 4 Minatar tasks).
> > >
> > > Finally, we reverse engineered the neural network-based LES into a “white-box” discovered ES, which can be interpreted and implements a type of truncation recombination (see Figure 5). The recombination weights are monotonic in the rank of the population members, which provides a loose theoretical justification. To the best of our knowledge, the discovered functional form (equation 4) was previously unknown and highlights the benefits of discovering new ES via meta-learning. Our work is therefore a novel investigation, which is significant not only   from an empirical perspective, but provides a clear direction for further theoretical investigations.
> > >
> > > We hope to have addressed the reviewer's remaining concerns and thank them for their time.
> > >
> > > Best wishes,
> > > The authors
> > >
> > > References:
> > >
> > > Agarwal, Rishabh, et al. “Deep reinforcement learning at the edge of the statistical precipice.” Advances in neural information processing systems 34 (2021): 29304-29320.

---

> ### Author Response · Authors · 2022-12-05
> **Request for rebuttal assessment/feedback**
>
> Dear reviewer Fevx,
> We again thank you for your time and feedback. Given that the ICLR discussion period will come to an end on December 12th, we kindly want to ask whether you could assess our rebuttal? Furthermore, we want to again highlight the extensive revision of the paper (e.g. clarifications in the related work section) and the accompanying additional experiments (2 more baselines, additional CNN neuroevolution experiments on MinAtar tasks and several ablations/meta-training comparisons in the SI).
>
> We hope that these additions have addressed your concerns with respect to the generalization capabilities of the meta-learned evolution strategy.
>
> Please let us know if any further questions may have come up.
> Have a good week,
> The authors.

---

### Author Response · Authors · 2022-11-13
**General comments to all reviewers**

We thank all reviewers for their thoughtful feedback. The manuscript has been significantly enhanced in order to address the points of concern. We want to quickly address the point of novelty and highlight the major revision additions to our paper:

**Novelty of our work**: A couple of reviewers have expressed concerns with regards to the novelty of our study. We are definitely not the first to propose the meta-learning of inductive biases in the context of optimization. Nonetheless, we are the first to show that meta-learned ES can achieve SotA performance on challenging high-dimensional neuroevolution tasks. This success has mainly been driven by our proposed self-attention-based parametrization of the ES update rule. After meta-training the LES is capable of strong generalization to different population sizes, problem dimensionalities and time horizons. This type of extreme generalization of meta-learned BBO has not been shown in any previous work. We have updated the related work section in order to highlight these contributions more clearly.

**Update to LES learning rate modulation**: We observed a form of time horizon overfitting during meta-testing (e.g. on the ur5e task) and have updated the learning rate modulation from an LSTM outputting per-parameter learning rates to a MLP. The processed inputs remain the same and we additionally chose to sample a starting timestep (t) during meta-training. This yielded improvements on a couple of neuroevolution tasks and improved scaling to larger population sizes. All figures have been updated and the discovered ES has changed from a linear equation mapping centered ranks to logits to an inverted sigmoidal relation. All of our qualitative results remain the same: LES provides a SotA meta-learned neuroevolution method.

**Additional revision experiments**: In order to further strengthen all of our claims, we have added a substantial amount of new experiments.
- *Baselines*: We added two more tuned baselines (PGPE and ASEBO) as requested by the reviewers. The proposed LES still robustly beats all the baselines on the majority of the considered neuroevolution tasks. The detailed learning curves can be found in SI Figure 14.
- *Additional tasks*: We provide an additional set of experiments on 4 MinAtar (Visual RL) tasks in SI Figure 19. LES beats OpenES and PGPE on 3 out of the 4 tasks and thereby also performs well on neuroevolution tasks, which parametrize agent policies with CNNs.
- *Comparing multiple different meta-trained LES*: We explored how much the discovered LES differ between meta-training runs. In SI Figure 17 we show that all of them are capable of effective neuroevolution. Still there remains some heterogeneity, which may open up exciting future work on ensembling different expert LES.
- *LES meta-training settings*: In SI Figures 20 and 21 (top) we investigated the effect of the meta-training settings (T, N, D). We show that the outcome of the MetaBBO procedure is largely robust to all three, but there are two exceptions: Too small population sizes and too long inner loop unrolls can decrease performance of the trained LES.
- *LES model size comparison*:  In SI Figure 21 (bottom) we compare different LES attention capacities and show that all considered model sizes tend to perform equally well. We therefore choose the smallest model for fast and memory efficient deployment.

Finally, we added more detail on the computational requirements of the meta-training procedure and the downstream evaluation protocol in the SI. All main text changes have been highlighted in red. We further provide detailed responses to all individual reviewers.

Again, we thank the reviewers for significantly improving the work and hope that these revisions could address your concerns. We are looking forward to your responses.

---

### Decision · Program_Chairs · 2023-01-20

**Decision:**

Accept: poster

**Justification For Why Not Higher Score:**

This is a nice use of meta-learning where the heuristic components in the optimization are replaced by a learning technique. However, there are a few inherent limitations, which are also outlined in the paper. Thus, the poster presentation is a right place.

**Justification For Why Not Lower Score:**

The authors did an excellent job in responding to individual comments. Most of reviewers were satisfied with the author response, increasing the overall score.

**Metareview: Summary, Strengths And Weaknesses:**

This paper addresses the black-box optimization and presents a method for discovering effective update rules for evolution strategy via meta-learning. The main idea is to parameterize a search strategy by self-attention-based architecture such that the update rule is invariant to the ordering of the candidate solutions. The heuristic components in the Gaussian ES is replaced by a meta-learning technique. This might be a general idea of meta-learning but the current work formulates it in a proper way and demonstrates its validity. Initial reviews pointed out several weaknesses and limitations, where details can be found in individual comments. Authors did an excellent job to resolve most of concerns in their rebuttal. During the discussion period, most of reviewers were satisfied with the author response, increasing the overall score. Some weaknesses, which are also outlined in the paper, includes the scaling self-attention mechanism, manually engineered features F_t, diagonal Gaussian ES. Nonetheless, most of reviewers agree that the paper contains interesting contributions, which deserve to be published. Last minute, I had a personal discussion with the reviewer who pointed out critical comments on lack of theoretical treatments and achieving diversity in meta-task distributions. With these extensive discussions with reviewers and thanks to the authors’ effort in responding to each comment, all reviewers finally agree to champion this paper.


**Note From Pc:**

if the above contains the word "oral" or "spotlight" please see: "oral" presentation means -> notable-top-5% and "spotlight" means -> notable-top-25%. As stated in our emails, we are disassociating presentation type from AC recommendations

**Summary Of Ac-Reviewer Meeting:**

Although I did organize a virtual meeting with all reviewers, I had a personal discussion with a particular reviewer and had sufficient discussion with the rest of reviewers. All these discussions lead to the consensus.